# Prethermalization of density-density correlations after an interaction quench in the Hubbard model

M. Kreye[1*], S. Kehrein[1]

**1** Institute for Theoretical Physics, University of Göttingen, Germany
* manuel.kreye@theorie.physik.uni-goettingen.de

August 29, 2019

## Abstract

**In weakly perturbed systems that are close to integrability, thermalization can be delayed by the formation of prethermalization plateaus. We study the build-up of density-density correlations after a weak interaction quench in the Hubbard model in $d > 1$ dimensions using unitary perturbation theory. Starting from a pre-quench state at temperature $T$, we show that the prethermalization values of the post-quench correlations are equal to the equilibrium values of the interacting model at the same temperature $T$. This is explained by the local character of density-density correlations.**

# 1  Introduction

## 1.1  Motivation

Seminal experiments with ultracold atoms have made it possible to study the thermalization dynamics of isolated quantum many-body systems out of equilibrium [1–4]. The high controllability of cold atoms in optical lattices allows for the simulation of artificial models and the implementation of quantum quenches, where in the subsequent nonequilibrium dynamics individual atoms can be tracked site- and time-resolved. For example, Kinoshita et al. demonstrated that a one-dimensional Bose gas brought out of equilibrium remains in a nonthermal steady state because of the integrability of the underlying model [2].

These experiments have stimulated theoretical research on the question how isolated quantum systems thermalize [5]. A pure state in an isolated system can be described by a density operator $\rho$ with $\mathrm{Tr}[\rho^2] = 1$. But, as it is subject to unitary time evolution, it can never evolve into a mixed thermal state with $\mathrm{Tr}[\rho^2] < 1$. However, for certain subsets of observables, a time-evolved pure state can become indistinguishable from a thermal state. The general view is that for local observables the environment acts as a thermal bath.

While after a quantum quench we expect generic nonintegrable systems to thermalize [3, 6], integrable systems, like in the experiment by Kinoshita et al. [2], usually do not thermalize because the set of conserved quantities strongly restricts the dynamics. However, the nonthermal steady states of integrable systems can be described by a generalized Gibbs ensemble (GGE) [7]. The natural question arises what happens to weakly perturbed systems, i.e., to nonintegrable systems that come close to integrability. Here, one often faces the phenomenon of prethermalization.

Prethermalization was discussed by Berges, Borsányi and Wetterich in the context of heavy-ion collisions [8]. They argued that in far-from-equilibrium settings there can be an intermediate time scale where bulk quantities, like the equation of state for hydrodynamical considerations, have already reached their equilibrium value while momentum-dependent mode quantities are still far from thermalization. In condensed matter physics, this concept was captured by Moeckel and Kehrein, who studied the thermalization dynamics of the momentum distribution function after a weak interaction quench in the Hubbard model using unitary perturbation theory [9]. They identified a prethermalization plateau where the momentum distribution becomes quasi-stationary but still differs from equilibrium. Their result was verified by numerical calculations in dynamical mean-field theory (DMFT) [10].

The relation between the prethermalization time scale and the perturbation strength in the Hubbard model led to the picture of near-integrability induced bottlenecks in the thermaliza-

tion dynamics emphasized by Kollar, Wolf and Eckstein [11]. They argued that prethermalization plateaus can also be predicted by generalized Gibbs ensembles and that nonthermal steady states in integrable systems can be interpreted as infinitely delayed prethermalization plateaus.

Meanwhile, prethermalization has become a topic of vast research interest. It has been studied e.g. in the Hubbard model [12–14], the three-dimensional Heisenberg model [15], one-dimensional spin chains [16, 17], the Luttinger model [18, 19], as well as in models with long-range interactions [20] and periodic driving [21]. All these works hint at the universality of prethermalization in perturbed systems [22, 23]. Experimentally, prethermalization has been observed in cold atom systems, e.g. in one-dimensional Bose gases [24, 25] and long-range interacting spin chains [26]. Furthermore, prethermalization has also been discussed in the context of quantum information [27], Anderson localization [28], many-body localization [29], quantum time crystals [30, 31] and the preheating of the early universe [32, 33].

In this paper, we extend the work of Moeckel and Kehrein and study a local quantity, more precisely the equal-time density-density correlation function, in the nonequilibrium Hubbard model in $d > 1$ dimensions. We consider a weak interaction quench, allowing us to use unitary perturbation theory, a method that avoids secular terms [34] and is especially suited to directly compare prethermalization to equilibrium values [9].

The pre-quench state has a temperature $T$ and reaches a prethermalized post-quench state where the correlation functions are equal to the equilibrium values of the interacting model at the same temperature $T$. Heating effects from the quench that would increase the temperature of the post-quench state will only show on a much longer time scale that is not covered by our approach.

## 1.2 Model

We study the real-time evolution in the Fermi-Hubbard model [35] in $d > 1$ dimensions,

$$\boldsymbol{H} = \sum_{k,\sigma} \epsilon_k : \boldsymbol{c}_{k\sigma}^\dagger \boldsymbol{c}_{k\sigma} : + \frac{U}{\Omega} \sum_{k_i',k_i} : \boldsymbol{c}_{k_1'\uparrow}^\dagger \boldsymbol{c}_{k_1\uparrow} \boldsymbol{c}_{k_2'\downarrow}^\dagger \boldsymbol{c}_{k_2\downarrow} : \delta_{k_1'+k_2',k_1+k_2} , \tag{1}$$

with a general dispersion relation $\epsilon_k$ and where $\epsilon_{\mathrm{F}} = 0$ is the Fermi energy. $U$ denotes the interaction strength, $\Omega$ the number of lattice sites, $\sigma \in \{\uparrow, \downarrow\}$ the spins and $k \in [-\pi, \pi]^d$ the momenta corresponding to reciprocal lattice vectors. For technical reasons, we use normal-ordering $: \cdot :$ with respect to the Gibbs state of the non-interacting Hamiltonian $\boldsymbol{H}_0 = \sum_{k,\sigma} \epsilon_k \boldsymbol{c}_{k\sigma}^\dagger \boldsymbol{c}_{k\sigma}$.

We implement a weak interaction quench by preparing the system in the ground state $|\psi_0\rangle$ of $\boldsymbol{H}_0$ and switching on the interaction to some finite value of $U$ at time $t = 0$. As the interaction $U$ is considered weak, we can treat the real-time evolution problem perturbatively.

The described quench setup and its perturbative treatment was studied by Moeckel and Kehrein, who calculated the time evolution of the momentum distribution function [9]. We build up our considerations from their work and expand it to include the real-time dynamics of the equal-time connected density-density correlation function

$$C_{x',x}^{\sigma'\sigma}(t) = \langle \boldsymbol{n}_{x',\sigma'}(t) \boldsymbol{n}_{x,\sigma}(t) \rangle - \langle \boldsymbol{n}_{x',\sigma'}(t) \rangle \langle \boldsymbol{n}_{x,\sigma}(t) \rangle , \tag{2}$$

where $\boldsymbol{n}_{x,\sigma}(t) = \Omega^{-1} \sum_{k',k} e^{i(k'-k)x} \boldsymbol{c}_{k'\sigma}^\dagger(t) \boldsymbol{c}_{k\sigma}(t)$ is the local density operator for spin-$\sigma$ particles at lattice site $x$.

# 2 Real-time evolution of the annihilation operator

The general idea is to solve the Heisenberg equation of motion for the annihilation operator $\boldsymbol{c}_{k\uparrow}(t)$ in the Hubbard model using unitary perturbation theory. We will calculate the perturbative expansion of $\boldsymbol{c}_{k\uparrow}(t)$ up to second order in $U$. This result can in principle be used for the construction of a wide class of observables. As an example, we will calculate density-density correlations, which can be evaluated for different initial states.

## 2.1 Unitary perturbation theory

A problem that often occurs in naive perturbative treatments of the Heisenberg equations of motion is the appearance of secular terms that grow with some power law in time. These secular terms emerge when the expansion in the small parameter indirectly includes an expansion in time.

In classical mechanics, one can avoid this problem by using canonical transformations that bring the Hamiltonian to normal-form, before one deals with the time evolution. Hackl and Kehrein extended this idea to the realm of quantum mechanics, where the canonical transformations must be replaced by unitary transformations [34].

The general scheme is depicted in Fig. 1: By (continuous) unitary transformations $\boldsymbol{U}$ one approximately diagonalizes the Hamiltonian $\boldsymbol{H}$ and transforms the observables $\boldsymbol{O}$ accordingly. In the energy-diagonal basis (denoted by a tilde), the Heisenberg equations of motion for the observables can be solved without the appearance of secular terms. After a backward transformation $\boldsymbol{U}^\dagger$ of the time-evolved observable to the original basis, one can calculate expectation values with respect to a given state $|\psi\rangle$.

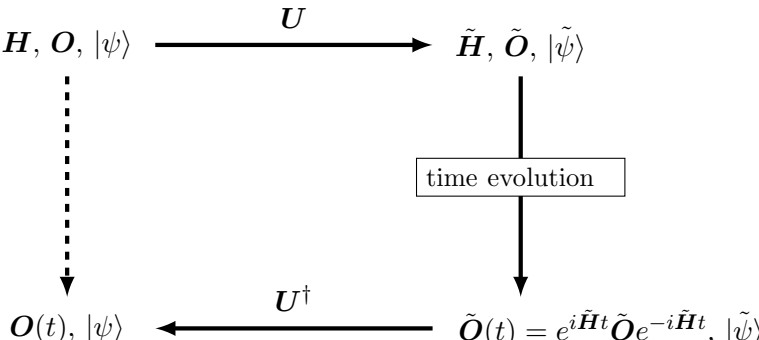

Figure 1: Illustration of the unitary perturbation theory scheme

Just like in the classical analogue, this forward-backward scheme can also be carried out perturbatively and still no secular terms will appear.

### The flow equation method

A method for approximately diagonalizing many-body Hamiltonians has been proposed by Wegner [36] and independently in the context of high-energy physics by Głazek and Wilson [37]. One applies a sequence of continuous unitary transformations defined by the flow

equation

$$\frac{\mathrm{d}\boldsymbol{H}(B)}{\mathrm{d}B} = [\boldsymbol{\eta}(B), \boldsymbol{H}(B)]_- \ , \tag{3}$$

where $\boldsymbol{H}(B=0)$ is the initial interacting Hamiltonian. Wegner showed that under rather general conditions the canonical generator

$$\boldsymbol{\eta}_{\mathrm{can.}}(B) \overset{\mathrm{def}}{=} [\boldsymbol{H}_0(B), \boldsymbol{H}_{\mathrm{int}}(B)]_- \ , \tag{4}$$

will effectively diagonalize the Hamiltonian in the limit $B \to \infty$, apart from degeneracies. Here, $\boldsymbol{H}_0(B)$ is the diagonal part of the Hamiltonian and $\boldsymbol{H}_{\mathrm{int}}(B)$ the interaction part. In the one-dimensional Hubbard model, $\boldsymbol{H}_0$ and $\boldsymbol{H}_{\mathrm{int}}$ would already commute at $B = 0$ and the flow would become featureless. Therefore, we have to assume $d > 1$.

The coupling between eq. (3) and (4) usually leads to the generation of an infinite series of higher-order interaction terms. This problem can be avoided by systematic expansions in the coupling parameter.

While the Hamiltonian will have a simple structure in the energy-diagonal basis, the complicated dynamics of the interacting system is shifted to the observables, which transform under

$$\frac{\mathrm{d}\boldsymbol{O}(B)}{\mathrm{d}B} = [\boldsymbol{\eta}(B), \boldsymbol{O}(B)]_- \tag{5}$$

and will hence become more intricate.

## Calculations in equilibrium

The forward-backward scheme in Fig. 1 is especially suited for our quench setup, because the initial state $|\psi_0\rangle$ is very simple and evaluations of expectation values after the backward transformation can be done by utilizing

$$\langle\psi_0|\boldsymbol{c}_{k\sigma}^\dagger \boldsymbol{c}_{k\sigma}|\psi_0\rangle = n_k \equiv \Theta(-\epsilon_k) \qquad (\text{for } \sigma = \uparrow, \downarrow) \ , \tag{6}$$

where the Fermi-Dirac distribution at zero temperature is just the Heaviside step function $\Theta(-\epsilon_k)$.

If we want to calculate equilibrium quantities of the interacting system, this can better be done in the energy-diagonal basis at $B = \infty$. This is because the ground state of an interacting Hamiltonian is more complicated, but will show the simple feature of eq. (6) in a basis, where the Hamiltonian is diagonal.

For our quench setup, we will use the flow equation results for $\tilde{\boldsymbol{O}}$ and $\boldsymbol{O}(t)$ to directly compare the equilibrium to the nonequilibrium setting.

## 2.2 Transformation of the Hamiltonian

As a second-order ansatz for the flowing Hamiltonian we choose

$$
\boldsymbol{H}(B) = \sum_{k,\sigma} \epsilon_k(B) : \boldsymbol{c}_{k\sigma}^\dagger \boldsymbol{c}_{k\sigma} : + \frac{1}{\Omega} \sum_{k_i',k_i} U_{k_1',k_1,k_2',k_2}(B) : \boldsymbol{c}_{k_1'\uparrow}^\dagger \boldsymbol{c}_{k_1\uparrow} \boldsymbol{c}_{k_2'\downarrow}^\dagger \boldsymbol{c}_{k_2\downarrow} : \delta_{k_1'+k_2',k_1+k_2}
$$

$$
+ \frac{1}{\Omega} \sum_{k_i',k_i} \sum_\sigma V_{k_1',k_1,k_2',k_2}(B) : \boldsymbol{c}_{k_1'\sigma}^\dagger \boldsymbol{c}_{k_1\sigma} \boldsymbol{c}_{k_2'\sigma}^\dagger \boldsymbol{c}_{k_2\sigma} : \delta_{k_1'+k_2',k_1+k_2}
$$

$$
+ \text{ higher-order interaction terms}
$$

$$
+ \mathcal{O}(U^3) \,, \tag{7}
$$

with initial values $\epsilon_k(B = 0) = \epsilon_k$, $U_{k_1',k_1,k_2',k_2}(B = 0) = U$ and $V_{k_1',k_1,k_2',k_2}(B = 0) = 0$. The higher-order interaction terms are not relevant for our calculation. Hence, the canonical generator from eq. (4) becomes

$$
\boldsymbol{\eta}(B) = \frac{1}{\Omega} \sum_{k_i',k_i} \Delta\epsilon_{k_1',k_1,k_2',k_2}(B) U_{k_1',k_1,k_2',k_2}(B) : \boldsymbol{c}_{k_1'\uparrow}^\dagger \boldsymbol{c}_{k_1\uparrow} \boldsymbol{c}_{k_2'\downarrow}^\dagger \boldsymbol{c}_{k_2\downarrow} : \delta_{k_1'+k_2',k_1+k_2}
$$

$$
+ \frac{1}{\Omega} \sum_{k_i',k_i} \sum_\sigma \Delta\epsilon_{k_1',k_1,k_2',k_2}(B) V_{k_1',k_1,k_2',k_2}(B) : \boldsymbol{c}_{k_1'\sigma}^\dagger \boldsymbol{c}_{k_1\sigma} \boldsymbol{c}_{k_2'\sigma}^\dagger \boldsymbol{c}_{k_2\sigma} : \delta_{k_1'+k_2',k_1+k_2}
$$

$$
+ \text{ higher-order interaction terms}
$$

$$
+ \mathcal{O}(U^3) \,, \tag{8}
$$

with $\Delta\epsilon_{k_1',k_1,k_2',k_2} \overset{\text{def}}{=} \epsilon_{k_1'} - \epsilon_{k_1} + \epsilon_{k_2'} - \epsilon_{k_2}$. With this generator, the flow equation for the Hamiltonian, given in eq. (3), yields

$$
\epsilon_k(B) = \epsilon_k + \frac{U^2}{\Omega^2} \sum_{k_1,k_2',k_2} \frac{1 - e^{-2(\Delta\epsilon_{k,k_1,k_2',k_2})^2 B}}{\Delta\epsilon_{k,k_1,k_2',k_2}} \left( (1 - n_{k_1})n_{k_2'}(1 - n_{k_2}) + n_{k_1}(1 - n_{k_2'})n_{k_2} \right)
$$

$$
+ \mathcal{O}(U^3) \,, \tag{9}
$$

$$
U_{k_1',k_1,k_2',k_2}(B) = U e^{-(\Delta\epsilon_{k_1',k_1,k_2',k_2})^2 B}
$$

$$
+ \mathcal{O}(U^2) \,, \tag{10}
$$

$$
V_{k_1',k_1,k_2',k_2}(B) = -\frac{U^2}{\Omega} \sum_{k_3',k_3} \Delta\epsilon_{k_1',k_1,k_3',k_3} \frac{e^{-(\Delta\epsilon_{k_1',k_1,k_3',k_3})^2 B} e^{-(\Delta\epsilon_{k_2',k_2,k_3,k_3'})^2 B} - e^{-(\Delta\epsilon_{k_1',k_1,k_2',k_2})^2 B}}{(\Delta\epsilon_{k_1',k_1,k_3',k_3})^2 + (\Delta\epsilon_{k_2',k_2,k_3,k_3'})^2 - (\Delta\epsilon_{k_1',k_1,k_2',k_2})^2}
$$

$$
\times (n_{k_3'} - n_{k_3})\delta_{k_3'+k_2,k_3+k_2'}
$$

$$
+ \mathcal{O}(U^3) \,. \tag{11}
$$

Clearly, the off-diagonal terms are exponentially surpressed throughout the flow. At $B = \infty$, only elastic collision terms with $\Delta\epsilon_{k_1',k_1,k_2',k_2} = 0$ survive. In this basis, the Hamiltonian takes on the form

$$
\tilde{\boldsymbol{H}} = \sum_{k,\sigma} \tilde{\epsilon}_k : \boldsymbol{c}_{k\sigma}^\dagger \boldsymbol{c}_{k\sigma} : + \frac{U}{\Omega} \sum_{k_i',k_i} : \boldsymbol{c}_{k_1'\uparrow}^\dagger \boldsymbol{c}_{k_1\uparrow} \boldsymbol{c}_{k_2'\downarrow}^\dagger \boldsymbol{c}_{k_2\downarrow} : \delta_{\epsilon_{k_1'}+\epsilon_{k_2'},\epsilon_{k_1}+\epsilon_{k_2}} \delta_{k_1'+k_2',k_1+k_2} + \mathcal{O}(U^2) \,, \tag{12}
$$

with a renormalized one-particle energy

$$\tilde{\epsilon}_k = \epsilon_k + \frac{U^2}{\Omega^2} \sum_{k_1,k_2',k_2} \frac{1}{\Delta\epsilon_{k,k_1,k_2',k_2}} \big((1-n_{k_1})n_{k_2'}(1-n_{k_2}) + n_{k_1}(1-n_{k_2'})n_{k_2}\big) + \mathcal{O}(U^3) \,. \quad (13)$$

The elastic collision terms in eq. (12) are exactly the contributions that appear in the quantum Boltzmann equation, from which we know to become relevant at time scales $t \sim \rho_{\mathrm{F}}^{-3}U^{-4}$ [38]. Our calculation, as we will show in sec. 2.4, is only stable for time scales up to and including $t \sim \rho_{\mathrm{F}}^{-1}U^{-2}$ and hence we neglect the elastic collisions.

## 2.3 Transformation of the annihilation operator

The annihilation operator is transformed under the flow equation from eq. (5),

$$\frac{\mathrm{d}\boldsymbol{c}_{k\uparrow}(B)}{\mathrm{d}B} = [\boldsymbol{\eta}(B), \boldsymbol{c}_{k\uparrow}(B)]_- \,. \quad (14)$$

The generator from eq. (8) causes a second-order flow to the following structure,

$$\begin{aligned}
\boldsymbol{c}_{k\uparrow}(B) = \; & h_k(B) : \boldsymbol{c}_{k\uparrow} : \\
& + \sum_{k_i',k_i} F_{k,k_1,k_2',k_2}(B) : \boldsymbol{c}_{k_1\uparrow}\boldsymbol{c}_{k_2'\downarrow}^\dagger\boldsymbol{c}_{k_2\downarrow} : \delta_{k+k_2',k_1+k_2} \\
& + \sum_{k_i',k_i} G_{k,k_1,k_2',k_2}(B) : \boldsymbol{c}_{k_1\uparrow}\boldsymbol{c}_{k_2'\uparrow}^\dagger\boldsymbol{c}_{k_2\uparrow} : \delta_{k+k_2',k_1+k_2} \\
& + \text{higher-order interaction terms} \\
& + \mathcal{O}(U^3) \,.
\end{aligned} \quad (15)$$

We will see in a moment that $F_{k,k_1,k_2',k_2}(B)$ only contributes to the correlation functions with its first-order correction. Hence, we only consider the following effective flow equations for the coefficients,

$$\begin{aligned}
\frac{\mathrm{d}h_k(B)}{\mathrm{d}B} = \; & \frac{U}{\Omega} \sum_{k_1,k_2',k_2} \Delta\epsilon_{k,k_1,k_2',k_2} e^{-(\Delta\epsilon_{k,k_1,k_2',k_2})^2 B} F_{k,k_1,k_2',k_2}(B) \\
& \times \big((1-n_{k_1})n_{k_2'}(1-n_{k_2}) + n_{k_1}(1-n_{k_2'})n_{k_2}\big)\delta_{k+k_2',k_1+k_2} \\
& + \mathcal{O}(U^3) \,, \quad (16) \\
\frac{\mathrm{d}F_{k,k_1,k_2',k_2}(B)}{\mathrm{d}B} = \; & -\frac{U}{\Omega}\Delta\epsilon_{k,k_1,k_2',k_2} e^{-(\Delta\epsilon_{k,k_1,k_2',k_2})^2 B} h_k(B) \\
& + \mathcal{O}(U^2) \,, \quad (17)
\end{aligned}$$

$$\frac{\mathrm{d}G_{k,k_1,k_2',k_2}(B)}{\mathrm{d}B} = \frac{U}{\Omega} \sum_{k_3',k_3} \Delta\epsilon_{k_3',k_3,k_2',k_2} e^{-(\Delta\epsilon_{k_3',k_3,k_2',k_2})^2 B} F_{k,k_1,k_3,k_3'}(B)$$

$$\times (n_{k_3'} - n_{k_3})\delta_{k_2'+k_3',k_2+k_3}$$

$$+ \frac{U^2}{\Omega^2} \sum_{k_3',k_3} (\Delta\epsilon_{k,k_1,k_2',k_2})(\Delta\epsilon_{k,k_1,k_3',k_3} - \Delta\epsilon_{k_2',k_2,k_3,k_3'}) h_k(B)$$

$$\times \frac{e^{-(\Delta\epsilon_{k,k_1,k_3',k_3})^2 B} e^{-(\Delta\epsilon_{k_2',k_2,k_3,k_3'})^2 B} - e^{-(\Delta\epsilon_{k,k_1,k_2',k_2})^2 B}}{(\Delta\epsilon_{k,k_1,k_3',k_3})^2 + (\Delta\epsilon_{k_2',k_2,k_3,k_3'})^2 - (\Delta\epsilon_{k,k_1,k_2',k_2})^2}$$

$$\times (n_{k_3'} - n_{k_3})\delta_{k_3'+k_2,k_3+k_2'}$$

$$+ \mathcal{O}(U^3) \,. \tag{18}$$

The perturbative solutions with the initial condition $\boldsymbol{c}_{k\uparrow}(B=0) =: \boldsymbol{c}_{k\uparrow}$ : are easily found,

$$h_k(B) = 1 - \frac{U^2}{2\Omega^2} \sum_{k_1,k_2',k_2} \left( \frac{1 - e^{-(\Delta\epsilon_{k,k_1,k_2',k_2})^2 B}}{\Delta\epsilon_{k,k_1,k_2',k_2}} \right)^2$$

$$\times \left( (1-n_{k_1})n_{k_2'}(1-n_{k_2}) + n_{k_1}(1-n_{k_2'})n_{k_2} \right)\delta_{k+k_2',k_1+k_2}$$

$$+ \mathcal{O}(U^3) \,, \tag{19}$$

$$F_{k,k_1,k_2',k_2}(B) = -\frac{U}{\Omega} \frac{1 - e^{-(\Delta\epsilon_{k,k_1,k_2',k_2})^2 B}}{\Delta\epsilon_{k,k_1,k_2',k_2}}$$

$$+ \mathcal{O}(U^2) \,, \tag{20}$$

$$G_{k,k_1,k_2',k_2}(B) = \frac{U^2}{\Omega^2} \sum_{k_3',k_3} \frac{\Delta\epsilon_{k_3',k_3,k_2',k_2}}{\Delta\epsilon_{k,k_1,k_3,k_3'}}$$

$$\times \left( \frac{1 - e^{-(\Delta\epsilon_{k_3',k_3,k_2',k_2})^2 B} e^{-(\Delta\epsilon_{k,k_1,k_3,k_3'})^2 B}}{(\Delta\epsilon_{k_3',k_3,k_2',k_2})^2 + (\Delta\epsilon_{k,k_1,k_3,k_3'})^2} - \frac{1 - e^{-(\Delta\epsilon_{k_3',k_3,k_2',k_2})^2 B}}{(\Delta\epsilon_{k_3',k_3,k_2',k_2})^2} \right)$$

$$\times (n_{k_3'} - n_{k_3})\delta_{k_2'+k_3',k_2+k_3}$$

$$+ \frac{U^2}{\Omega^2} \sum_{k_3',k_3} \frac{(\Delta\epsilon_{k,k_1,k_2',k_2})(\Delta\epsilon_{k,k_1,k_3',k_3} - \Delta\epsilon_{k_2',k_2,k_3,k_3'})}{(\Delta\epsilon_{k,k_1,k_3',k_3})^2 + (\Delta\epsilon_{k_2',k_2,k_3,k_3'})^2 - (\Delta\epsilon_{k,k_1,k_2',k_2})^2}$$

$$\times \left( \frac{1 - e^{-(\Delta\epsilon_{k,k_1,k_3',k_3})^2 B} e^{-(\Delta\epsilon_{k_2',k_2,k_3,k_3'})^2 B}}{(\Delta\epsilon_{k,k_1,k_3',k_3})^2 + (\Delta\epsilon_{k_2',k_2,k_3,k_3'})^2} - \frac{1 - e^{-(\Delta\epsilon_{k,k_1,k_2',k_2})^2 B}}{(\Delta\epsilon_{k,k_1,k_2',k_2})^2} \right)$$

$$\times (n_{k_3'} - n_{k_3})\delta_{k_3'+k_2,k_3+k_2'}$$

$$+ \mathcal{O}(U^3) \,. \tag{21}$$

Taking the limit $B \to \infty$, we have a solution for $\tilde{\boldsymbol{c}}_{k\uparrow}$ in the energy-diagonal basis that we can use for equilibrium considerations. For the nonequilibrium quench setup, we now need to time-evolve the annihilation operator and then transform it back to the original basis at $B = 0$.

## 2.4 Time evolution and backward transformation

As pointed out in sec. (2.2), at $B = \infty$ the time evolution up to and including $t \sim \rho_{\mathrm{F}}^{-1} U^{-2}$ is simply governed by the quadratic Hamiltonian $\tilde{\boldsymbol{H}} = \sum_{k,\sigma} \tilde{\epsilon}_k : \boldsymbol{c}_{k\sigma}^{\dagger} \boldsymbol{c}_{k\sigma} :$. For the coefficients of the annihilation operator, we get

$$\tilde{h}_k(t) = e^{-i\tilde{\epsilon}_k t} \tilde{h}_k \ , \tag{22}$$

$$\tilde{F}_{k,k_1,k_2',k_2}(t) = e^{-i(\tilde{\epsilon}_{k_1} - \tilde{\epsilon}_{k_2'} + \tilde{\epsilon}_{k_2})t} \tilde{F}_{k,k_1,k_2',k_2} \ , \tag{23}$$

$$\tilde{G}_{k,k_1,k_2',k_2}(t) = e^{-i(\tilde{\epsilon}_{k_1} - \tilde{\epsilon}_{k_2'} + \tilde{\epsilon}_{k_2})t} \tilde{G}_{k,k_1,k_2',k_2} \ . \tag{24}$$

These time-evolved functions are now used as initial conditions for the backward transformation to $B = 0$ that is also given by eqs. (16) - (18). Integrating the flow equations backwards yields

$$h_k(t) = e^{-i\tilde{\epsilon}_k t}$$
$$- \frac{U^2}{\Omega^2} e^{-i\epsilon_k t} \sum_{k_1,k_2',k_2} \frac{1 - e^{i(\Delta\epsilon_{k,k_1,k_2',k_2})t}}{(\Delta\epsilon_{k,k_1,k_2',k_2})^2}$$
$$\times \left( (1 - n_{k_1}) n_{k_2'} (1 - n_{k_2}) + n_{k_1} (1 - n_{k_2'}) n_{k_2} \right) \delta_{k+k_2',k_1+k_2}$$
$$+ \mathcal{O}(U^3) \ , \tag{25}$$

$$F_{k,k_1,k_2',k_2}(t) = \frac{U}{\Omega} e^{-i\epsilon_k t} \frac{1 - e^{i(\Delta\epsilon_{k,k_1,k_2',k_2})t}}{\Delta\epsilon_{k,k_1,k_2',k_2}}$$
$$+ \mathcal{O}(U^2) \ , \tag{26}$$

$$G_{k,k_1,k_2',k_2}(t) = -\frac{U^2}{\Omega^2} e^{-i\epsilon_k t} \sum_{k_3',k_3} \frac{\Delta\epsilon_{k_3',k_3,k_2',k_2}}{\Delta\epsilon_{k,k_1,k_3,k_3'}}$$
$$\times \left( \frac{1 - e^{i(\Delta\epsilon_{k,k_1,k_2',k_2})t}}{(\Delta\epsilon_{k_3',k_3,k_2',k_2})^2 + (\Delta\epsilon_{k,k_1,k_3,k_3'})^2} - \frac{e^{i(\Delta\epsilon_{k,k_1,k_3,k_3'})t} - e^{i(\Delta\epsilon_{k,k_1,k_2',k_2})t}}{(\Delta\epsilon_{k_3',k_3,k_2',k_2})^2} \right)$$
$$\times (n_{k_3'} - n_{k_3}) \delta_{k_2'+k_3',k_2+k_3}$$
$$+ \frac{U^2}{\Omega^2} e^{-i\epsilon_k t} \sum_{k_3',k_3} \frac{(\Delta\epsilon_{k,k_1,k_2',k_2})(\Delta\epsilon_{k,k_1,k_3',k_3} - \Delta\epsilon_{k_2',k_2,k_3,k_3'})}{(\Delta\epsilon_{k,k_1,k_3',k_3})^2 + (\Delta\epsilon_{k_2',k_2,k_3,k_3'})^2 - (\Delta\epsilon_{k,k_1,k_2',k_2})^2}$$
$$\times \left( \frac{e^{i(\Delta\epsilon_{k,k_1,k_2',k_2})t} - 1}{(\Delta\epsilon_{k,k_1,k_3',k_3})^2 + (\Delta\epsilon_{k_2',k_2,k_3,k_3'})^2} - \frac{e^{i(\Delta\epsilon_{k,k_1,k_2',k_2})t} - 1}{(\Delta\epsilon_{k,k_1,k_2',k_2})^2} \right)$$
$$\times (n_{k_3'} - n_{k_3}) \delta_{k_3'+k_2,k_3+k_2'}$$
$$+ \mathcal{O}(U^3) \ . \tag{27}$$

**Time scales:** Moeckel and Kehrein argued that the perturbative solution is stable up to and including time scales $t \sim \rho_{\mathrm{F}}^{-1} U^{-2}$, where $\rho_{\mathrm{F}}$ is the density of states at the Fermi edge [9]. In order to see this, we evaluate the expression from eq. (25) introducing energy integrals,

$$h_k(t) = e^{-i\epsilon_k t} - U^2 e^{-i\epsilon_k t} \int_{-\infty}^{\infty} \mathrm{d}E \frac{1 - e^{i(\epsilon_k - E)t}}{(\epsilon_k - E)^2} I_k(E) + \mathcal{O}(U^3) \ . \tag{28}$$

At temperature $T$, the phase space factor $I_k(E)$ is $\propto \rho_F^3 \max\{E^2, T^2\}$. For zero temperature, the integral in eq. (28) converges for all times at the Fermi surface, where $\epsilon_k = \epsilon_F = 0$. Away from the Fermi surface, the integral diverges as $\sim \epsilon_k^2 t$ for $\epsilon_k \gtrsim T$ and as $\sim T^2 t$ for $\epsilon_k \lesssim T$. Therefore, the second order correction of $h_k(t)$ becomes comparable to 1 for times $t \sim \rho_F^{-3} U^{-2} \min\{\epsilon_k^{-2}, T^{-2}\}$. This implies that the perturbative nature of our approach is valid until times $t \lesssim \rho_F^{-1} U^{-2}$ for a worst case estimate where $\epsilon_k$ is of order the bandwidth. However, one often considers only dynamical contributions in the vicinity of the Fermi edge, $\epsilon_k \approx 0$, and at low temperature, which much improves the stability of the time evolution.

Together with the general structure of the annihilation operator from eq. (15), we have reached a perturbative solution of the Heisenberg equation of motion for this operator that can be used to construct a wide class of observables. Before we use this for the evaluation of density-density correlations, we check our result for consistency.

**Consistency check 1:** preservation of canonical anticommutation relation

As the sequence of forward transformation, time evolution and backward transformation is completely unitary, the canonical anticommutation relation

$$\left[ \boldsymbol{c}_{k\uparrow}(t), \boldsymbol{c}_{k'\uparrow}^\dagger(t) \right]_+ \overset{!}{=} \delta_{k,k'} + \mathcal{O}(U^3) \tag{29}$$

should be preserved, at least in a perturbative sense. This condition leads to a relation between $h_k(t)$, $F_{k,k_1,k_2',k_2}(t)$ and $G_{k,k_1,k_2',k_2}(t)$ that is indeed fulfilled by our solutions from eqs. (25) - (27), see App. A.

**Consistency check 2:** total spin-up particle number

From our perturbative solution for $\boldsymbol{c}_{k\uparrow}(t)$, we can easily calculate the operator for the to-tal spin-up particle number,

$$\boldsymbol{N}_\uparrow(t) \overset{\text{def}}{=} \sum_k \boldsymbol{c}_{k\uparrow}^\dagger(t) \boldsymbol{c}_{k\uparrow}(t) \, , \tag{30}$$

which must be conserved, because it commutes with the Hamiltonian. We can show that the above solutions are also consistent with this condition, see App. A.

## 3 Equal-time connected density-density correlation function

Now, we are able to evaluate expectation values of time-evolved observables with respect to the initial state $|\psi_0\rangle$, which corresponds to the nonequilibrium quench setup. If we calculate expectation values of observables in the basis at $B = \infty$ with respect to the same state $|\psi_0\rangle$, this will correspond to the interacting Hubbard model in equilibrium.
The quantity of interest is the equal-time connected density-density correlation function from

eq. (2),

$$C_{x',x}^{\sigma'\sigma}(t) = \frac{1}{\Omega^2} \sum_{k',k,q',q} e^{i(k'-k)x'} e^{i(q'-q)x} \langle \boldsymbol{c}_{k'\sigma'}^\dagger(t) \boldsymbol{c}_{k\sigma'}(t) \boldsymbol{c}_{q'\sigma}^\dagger(t) \boldsymbol{c}_{q\sigma}(t) \rangle$$

$$- \frac{1}{\Omega^2} \sum_{k',k,q',q} e^{i(k'-k)x'} e^{i(q'-q)x} \langle \boldsymbol{c}_{k'\sigma'}^\dagger(t) \boldsymbol{c}_{k\sigma'}(t) \rangle \langle \boldsymbol{c}_{q'\sigma}^\dagger(t) \boldsymbol{c}_{q\sigma}(t) \rangle . \tag{31}$$

We will distinguish to two cases of antiparallel-spin and parallel-spin correlations, where we have $C_{x',x}^{\uparrow\downarrow}(t) \equiv C_{x',x}^{\downarrow\uparrow}(t)$ and $C_{x',x}^{\uparrow\uparrow}(t) \equiv C_{x',x}^{\downarrow\downarrow}(t)$ due to the spin-symmetry of the Hubbard model. Details of the calculation can be found in App. B.

## 3.1 Correlations between antiparallel spins

For the case of antiparallel spins, the correlation function has a leading order contribution that is of first order in $U$. In the nonequilibrium quench scenario, we get the following correlation function,

$$C_{x',x}^{\uparrow\downarrow}(t) = \frac{2U}{\Omega^3} \sum_{k',k} e^{i(k'-k)(x'-x)}(n_{k'} - n_k) \sum_{q',q} \frac{1 - \cos\left((\Delta\epsilon_{k',k,q',q})t\right)}{\Delta\epsilon_{k',k,q',q}}(1 - n_{q'})n_q \delta_{k'+q',k+q}$$

$$+ \mathcal{O}(U^2) , \tag{32}$$

while for the interacting Hubbard model in equilibrium, we get

$$C_{x',x}^{\text{eq.}\uparrow\downarrow} = \frac{2U}{\Omega^3} \sum_{k',k} e^{i(k'-k)(x'-x)}(n_{k'} - n_k) \sum_{q',q} \frac{1}{\Delta\epsilon_{k',k,q',q}}(1 - n_{q'})n_q \delta_{k'+q',k+q}$$

$$+ \mathcal{O}(U^2) . \tag{33}$$

Now, we calculate the time average of the nonequilibrium correlation function,

$$\overline{C_{x',x}^{\uparrow\downarrow}(t)} \overset{\text{def}}{=} \lim_{t\to\infty} \frac{1}{t} \int_0^t \mathrm{d}t' C_{x',x}^{\uparrow\downarrow}(t') . \tag{34}$$

As our perturbative ansatz only covers time scales up to and including the prethermalization regime, this time average equals the prethermalization value of the correlation function though we integrate to $t = \infty$. The integration to $t = \infty$ also makes the initial transient of the correlation function play no role for the time average.

The time average of eq. (32), where only the cos-function drops out, is equal to the equilibrium result, at least in leading order. Hence, the prethermalization value is

$$C_{x',x}^{\text{pre.}\uparrow\downarrow} \equiv \overline{C_{x',x}^{\uparrow\downarrow}(t)}$$

$$= C_{x',x}^{\text{eq.}\uparrow\downarrow} + \mathcal{O}(U^2) . \tag{35}$$

## 3.2  Correlations between parallel spins

For parallel spins, the correlation function is of second order in $U$ and therefore more intricate. In the nonequilibrium setting, we find

$$
\begin{aligned}
C^{\uparrow\uparrow}_{x',x}(t) = {} & \frac{1}{\Omega^2} \sum_{k',k} e^{i(k'-k)(x'-x)} n_{k'}(1-n_k) \\
& - \frac{4U^2}{\Omega^4} \sum_{k',k} e^{i(k'-k)(x'-x)} n_{k'}(1-n_k) \\
& \qquad \times \sum_{k_1,k_2',k_2} \frac{1 - \cos\left((\Delta\epsilon_{k,k_1,k_2',k_2})t\right)}{(\Delta\epsilon_{k,k_1,k_2',k_2})^2} \\
& \qquad\quad \times \left((1-n_{k_1})n_{k_2'}(1-n_{k_2}) + n_{k_1}(1-n_{k_2'})n_{k_2}\right)\delta_{k+k_2',k_1+k_2} \\
& + \frac{2U^2}{\Omega^4} \sum_{k',k} e^{i(k'-k)(x'-x)} n_{k'}(1-n_k) \\
& \qquad \times \sum_{k_i',k_i} \frac{1 - \cos\left((\Delta\epsilon_{k',k_1',k_2',k_2})t\right) - \cos\left((\Delta\epsilon_{k,k_1,k_2',k_2})t\right) + \cos\left((\Delta\epsilon_{k',k,k_1,k_1'})t\right)}{(\Delta\epsilon_{k',k_1',k_2',k_2})(\Delta\epsilon_{k,k_1,k_2',k_2})} \\
& \qquad\quad \times n_{k_2'}(1-n_{k_2})\delta_{k+k_2',k_1+k_2}\delta_{k'+k_1,k+k_1'} \\
& - \frac{2U^2}{\Omega^4} \sum_{k',k} e^{i(k'-k)(x'-x)} n_{k'}(1-n_k) \\
& \qquad \times \sum_{k_i',k_i} \frac{\Delta\epsilon_{k',k,k_2',k_2}}{\Delta\epsilon_{k_1',k_1,k_2,k_2'}} \left( \frac{1 - \cos\left((\Delta\epsilon_{k',k,k_1',k_1})t\right)}{(\Delta\epsilon_{k',k,k_2',k_2})^2 + (\Delta\epsilon_{k_1',k_1,k_2,k_2'})^2} - \frac{1 - \cos\left((\Delta\epsilon_{k',k,k_2',k_2})t\right)}{(\Delta\epsilon_{k',k,k_2',k_2})^2} \right) \\
& \qquad\quad \times (n_{k_1'}-n_{k_1})(n_{k_2'}-n_{k_2})\delta_{k'+k_1',k+k_1}\delta_{k'+k_2',k+k_2} \\
& + \frac{2U^2}{\Omega^4} \sum_{k',k} e^{i(k'-k)(x'-x)} \left(n_{k'}(1-n_k) + (1-n_{k'})n_k\right) \\
& \qquad \times \sum_{k_i',k_i} \frac{\Delta\epsilon_{k,k_1,k_2',k_2}}{\Delta\epsilon_{k',k_1',k_2',k_2}} \left( \frac{1 - \cos\left((\Delta\epsilon_{k',k,k_1,k_1'})t\right)}{(\Delta\epsilon_{k',k_1',k_2',k_2})^2 + (\Delta\epsilon_{k,k_1,k_2',k_2})^2} - \frac{1 - \cos\left((\Delta\epsilon_{k,k_1,k_2',k_2})t\right)}{(\Delta\epsilon_{k,k_1,k_2',k_2})^2} \right) \\
& \qquad\quad \times n_{k_1}(n_{k_2'}-n_{k_2})\delta_{k+k_2',k_1+k_2}\delta_{k'+k_1,k+k_1'} \\
& + \frac{2U^2}{\Omega^4} \sum_{k',k} e^{i(k'-k)(x'-x)} n_{k'}(1-n_k) \\
& \qquad \times \sum_{k_i',k_i} \frac{(\Delta\epsilon_{k_1',k_1,k_2,k_2'}) + (\Delta\epsilon_{k',k,k_2,k_2'})}{(\Delta\epsilon_{k',k,k_1,k_1'})} \left( \frac{1 - \cos\left((\Delta\epsilon_{k',k,k_1,k_1'})t\right)}{(\Delta\epsilon_{k_1',k_1,k_2,k_2'})^2 + (\Delta\epsilon_{k',k,k_2,k_2'})^2} \right) \\
& \qquad\quad \times (n_{k_1'}-n_{k_1})(n_{k_2'}-n_{k_2})\delta_{k'+k_1,k+k_1'}\delta_{k'+k_2,k+k_2'}
\end{aligned}
$$

$$+ \frac{2U^2}{\Omega^4} \sum_{k',k} e^{i(k'-k)(x'-x)} \big(n_{k'}(1-n_k) + (1-n_{k'})n_k\big)$$

$$\times \sum_{k'_i,k_i} \frac{(\Delta\epsilon_{k',k'_1,k'_2,k_2}) + (\Delta\epsilon_{k,k_1,k'_2,k_2})}{(\Delta\epsilon_{k',k,k_1,k'_1})} \left( \frac{1 - \cos\big((\Delta\epsilon_{k',k,k_1,k'_1})t\big)}{(\Delta\epsilon_{k',k'_1,k'_2,k_2})^2 + (\Delta\epsilon_{k,k_1,k'_2,k_2})^2} \right)$$

$$\times n_{k'_1}(n_{k'_2} - n_{k_2})\delta_{k+k'_2,k_1+k_2}\delta_{k'+k_1,k+k'_1}$$

$$+ \frac{U^2}{\Omega^4} \sum_{k',k} e^{i(k'-k)(x'-x)} n_{k'}(1-n_k)$$

$$\times \sum_{k'_i,k_i} \frac{1 - \cos\big((\Delta\epsilon_{k',k,k'_1,k_1})t\big) - \cos\big((\Delta\epsilon_{k',k,k'_2,k_2})t\big) + \cos\big((\Delta\epsilon_{k'_1,k_1,k_2,k'_2})t\big)}{(\Delta\epsilon_{k',k,k'_1,k_1})(\Delta\epsilon_{k',k,k'_2,k_2})}$$

$$\times (n_{k'_1} - n_{k_1})(n_{k'_2} - n_{k_2})\delta_{k'+k'_1,k+k_1}\delta_{k'+k'_2,k+k_2}$$

$$- \frac{2U^2}{\Omega^4} \sum_{k',k} e^{i(k'-k)(x'-x)} n_{k'}$$

$$\times \sum_{k'_i,k_i} \frac{1 - \cos\big((\Delta\epsilon_{k',k'_1,k'_2,k_2})t\big) - \cos\big((\Delta\epsilon_{k,k_1,k'_2,k_2})t\big) + \cos\big((\Delta\epsilon_{k',k,k_1,k'_1})t\big)}{(\Delta\epsilon_{k',k'_1,k'_2,k_2})(\Delta\epsilon_{k,k_1,k'_2,k_2})}$$

$$\times (1 - n_{k_1})n_{k'_2}(1 - n_{k_2})\delta_{k+k'_2,k_1+k_2}\delta_{k'+k_1,k+k'_1}$$

$$+ \frac{4U^2}{\Omega^4} \sum_{k',k} e^{i(k'-k)(x'-x)} n_{k'}$$

$$\times \sum_{k_1,k'_2,k_2} \frac{1 - \cos\big((\Delta\epsilon_{k,k_1,k'_2,k_2})t\big)}{(\Delta\epsilon_{k,k_1,k'_2,k_2})^2}(1 - n_{k_1})n_{k'_2}(1 - n_{k_2})\delta_{k+k'_2,k_1+k_2}$$

$$+ \mathcal{O}(U^3) . \tag{36}$$

For this solution, a further consistency check is sensible.

**Consistency check 3:** variance of total spin-up particle number

As the total spin-up particle number is conserved, we expect this also for its variance. The variance is obtained by a lattice summation over the parallel-spin correlation function,

$$\sum_{x',x} C^{\uparrow\uparrow}_{x',x}(t) = \langle(\boldsymbol{N}_\uparrow(t))^2\rangle - (\langle\boldsymbol{N}_\uparrow(t)\rangle)^2 . \tag{37}$$

In App. A, we show that the lattice summation over our result from eq. (36) indeed yields the time-independent solution

$$\sum_{x',x} C^{\uparrow\uparrow}_{x',x}(t) = \sum_k n_k(1-n_k) + \mathcal{O}(U^3) . \tag{38}$$

For the interacting Hubbard model in equilibrium, the parallel-spin correlation function is

$$
C_{x',x}^{\text{eq.}\uparrow\uparrow} = \frac{1}{\Omega^2} \sum_{k',k} e^{i(k'-k)(x'-x)} n_{k'}(1-n_k)
$$

$$
- \frac{2U^2}{\Omega^4} \sum_{k',k} e^{i(k'-k)(x'-x)} n_{k'}(1-n_k)
$$

$$
\times \sum_{k_1,k_2',k_2} \frac{1}{(\Delta\epsilon_{k,k_1,k_2',k_2})^2} \big((1-n_{k_1})n_{k_2'}(1-n_{k_2}) + n_{k_1}(1-n_{k_2'})n_{k_2}\big)\delta_{k+k_2',k_1+k_2}
$$

$$
+ \frac{2U^2}{\Omega^4} \sum_{k',k} e^{i(k'-k)(x'-x)} n_{k'}(1-n_k)
$$

$$
\times \sum_{k_i',k_i} \frac{1}{(\Delta\epsilon_{k',k_1',k_2',k_2})(\Delta\epsilon_{k,k_1,k_2',k_2})} n_{k_2'}(1-n_{k_2})\delta_{k+k_2',k_1+k_2}\delta_{k'+k_1,k+k_1'}
$$

$$
- \frac{2U^2}{\Omega^4} \sum_{k',k} e^{i(k'-k)(x'-x)} n_{k'}(1-n_k)
$$

$$
\times \sum_{k_i',k_i} \frac{\Delta\epsilon_{k',k,k_2',k_2}}{\Delta\epsilon_{k_1',k_1,k_2,k_2'}} \left( \frac{1}{(\Delta\epsilon_{k',k,k_2',k_2})^2 + (\Delta\epsilon_{k_1',k_1,k_2,k_2'})^2} - \frac{1}{(\Delta\epsilon_{k',k,k_2',k_2})^2} \right)
$$

$$
\times (n_{k_1'} - n_{k_1})(n_{k_2'} - n_{k_2})\delta_{k'+k_1',k+k_1}\delta_{k'+k_2',k+k_2}
$$

$$
+ \frac{2U^2}{\Omega^4} \sum_{k',k} e^{i(k'-k)(x'-x)} \big(n_{k'}(1-n_k) + (1-n_{k'})n_k\big)
$$

$$
\times \sum_{k_i',k_i} \frac{\Delta\epsilon_{k,k_1,k_2',k_2}}{\Delta\epsilon_{k',k_1',k_2',k_2}} \left( \frac{1}{(\Delta\epsilon_{k',k_1',k_2',k_2})^2 + (\Delta\epsilon_{k,k_1,k_2',k_2})^2} - \frac{1}{(\Delta\epsilon_{k,k_1,k_2',k_2})^2} \right)
$$

$$
\times n_{k_1}(n_{k_2'} - n_{k_2})\delta_{k+k_2',k_1+k_2}\delta_{k'+k_1,k+k_1'}
$$

$$
+ \frac{2U^2}{\Omega^4} \sum_{k',k} e^{i(k'-k)(x'-x)} n_{k'}(1-n_k)
$$

$$
\times \sum_{k_i',k_i} \frac{(\Delta\epsilon_{k_1',k_1,k_2,k_2'}) + (\Delta\epsilon_{k',k,k_2,k_2'})}{(\Delta\epsilon_{k',k,k_1,k_1'})} \left( \frac{1}{(\Delta\epsilon_{k_1',k_1,k_2,k_2'})^2 + (\Delta\epsilon_{k',k,k_2,k_2'})^2} \right)
$$

$$
\times (n_{k_1'} - n_{k_1})(n_{k_2'} - n_{k_2})\delta_{k'+k_1,k+k_1'}\delta_{k'+k_2,k+k_2'}
$$

$$
+ \frac{2U^2}{\Omega^4} \sum_{k',k} e^{i(k'-k)(x'-x)} \big(n_{k'}(1-n_k) + (1-n_{k'})n_k\big)
$$

$$
\times \sum_{k_i',k_i} \frac{(\Delta\epsilon_{k',k_1',k_2',k_2}) + (\Delta\epsilon_{k,k_1,k_2',k_2})}{(\Delta\epsilon_{k',k,k_1,k_1'})} \left( \frac{1}{(\Delta\epsilon_{k',k_1',k_2',k_2})^2 + (\Delta\epsilon_{k,k_1,k_2',k_2})^2} \right)
$$

$$
\times n_{k_1'}(n_{k_2'} - n_{k_2})\delta_{k+k_2',k_1+k_2}\delta_{k'+k_1,k+k_1'}
$$

$$
+ \frac{U^2}{\Omega^4} \sum_{k',k} e^{i(k'-k)(x'-x)} n_{k'}(1-n_k)
$$

$$
\times \sum_{k_i',k_i} \frac{1}{(\Delta\epsilon_{k',k,k_1',k_1})(\Delta\epsilon_{k',k,k_2',k_2})} (n_{k_1'} - n_{k_1})(n_{k_2'} - n_{k_2})\delta_{k'+k_1',k+k_1}\delta_{k'+k_2',k+k_2}
$$

$$
-\frac{2U^2}{\Omega^4} \sum_{k',k} e^{i(k'-k)(x'-x)} n_{k'}
$$

$$
\times \sum_{k'_i,k_i} \frac{1}{(\Delta\epsilon_{k',k'_1,k'_2,k_2})(\Delta\epsilon_{k,k_1,k'_2,k_2})}(1-n_{k_1})n_{k'_2}(1-n_{k_2})\delta_{k+k'_2,k_1+k_2}\delta_{k'+k_1,k+k'_1}
$$

$$
+\frac{2U^2}{\Omega^4} \sum_{k',k} e^{i(k'-k)(x'-x)} n_{k'}
$$

$$
\times \sum_{k_1,k'_2,k_2} \frac{1}{(\Delta\epsilon_{k,k_1,k'_2,k_2})^2}(1-n_{k_1})n_{k'_2}(1-n_{k_2})\delta_{k+k'_2,k_1+k_2}
$$

$$
+\mathcal{O}(U^3) . \tag{39}
$$

Comparing this to the nonequilibrium result, we realize that there are different prefactors in different terms. This hampers a direct relation of the prethermalization value of the post-quench state to the equilibrium value of the interacting model. We remark that the difference is due to the second-order corrections of $h_k(t)$. These have also been responsible for the deviation of the nonequilibrium momentum distribution function from the equilibrium value in the calculation by Moeckel and Kehrein [9].

In the following, we will show that the prethermalization value of the parallel-spin correlation function is equal to the equilibrium value at least for small momentum transfer, i.e., up to linear order in $q$, where $q$ is the momentum in Fourier space associated with the distance $x'-x$ in real space.

In order to perform a small momentum expansion in Fourier space, we need do apply the limit of infinite spatial dimensions.

## 3.3  Small $q$-limit of parallel-spin correlations in infinite spatial dimensions

We Fourier transform the parallel-spin correlation function to momentum space, $\hat{C}_q^{\uparrow\uparrow}(t) = \frac{1}{\Omega} \sum_{x'-x} e^{-iq(x'-x)} C_{x',x}^{\uparrow\uparrow}(t)$, take the limit of an infinite dimensional lattice [39], which allows us to replace sums over momenta by energy integrals, and expand the correlation function for small momentum $q$, which corresponds to the long-range behavior in real space.

For the nonequilibrium function, we find

$$
\hat{C}_q^{\uparrow\uparrow}(t) = \frac{1}{\Omega^2} \sum_k n_{k+q}(1 - n_k)
$$

$$
- \frac{2U^2}{\Omega^2} \sum_k n_{k+q}(1 - n_k) \int d\epsilon_{1'} \int d\epsilon_1 \int d\epsilon_{2'} \int d\epsilon_2 D(\epsilon_{1'})D(\epsilon_1)D(\epsilon_{2'})D(\epsilon_2)
$$

$$
\times \frac{\Delta\epsilon_{2',2}}{\Delta\epsilon_{1',1,2,2'}} \left( \frac{1 - \cos\left((\Delta\epsilon_{1',1})t\right)}{(\Delta\epsilon_{2',2})^2 + (\Delta\epsilon_{1',1,2,2'})^2} - \frac{1 - \cos\left((\Delta\epsilon_{2',2})t\right)}{(\Delta\epsilon_{2',2})^2} \right) (n_{1'} - n_1)(n_{2'} - n_2)
$$

$$
+ \frac{2U^2}{\Omega^2} \sum_k n_{k+q}(1 - n_k) \int d\epsilon_{1'} \int d\epsilon_1 \int d\epsilon_{2'} \int d\epsilon_2 D(\epsilon_{1'})D(\epsilon_1)D(\epsilon_{2'})D(\epsilon_2)
$$

$$
\times \frac{(\Delta\epsilon_{1',1,2,2'}) + (\Delta\epsilon_{2,2'})}{(\Delta\epsilon_{1,1'})} \left( \frac{1 - \cos\left((\Delta\epsilon_{1,1'})t\right)}{(\Delta\epsilon_{1',1,2,2'})^2 + (\Delta\epsilon_{2,2'})^2} \right) (n_{k_1'} - n_{k_1})(n_{k_2'} - n_{k_2})
$$

$$
+ \frac{U^2}{\Omega^2} \sum_k n_{k+q}(1 - n_k) \int d\epsilon_{1'} \int d\epsilon_1 \int d\epsilon_{2'} \int d\epsilon_2 D(\epsilon_{1'})D(\epsilon_1)D(\epsilon_{2'})D(\epsilon_2)
$$

$$
\times \frac{1 - \cos\left((\Delta\epsilon_{1',1})t\right) - \cos\left((\Delta\epsilon_{2',2})t\right) + \cos\left((\Delta\epsilon_{1',1,2,2'})t\right)}{(\Delta\epsilon_{1',1})(\Delta\epsilon_{2',2})}(n_{1'} - n_1)(n_{2'} - n_2)
$$

$$
+ \mathcal{O}(q^2) + \mathcal{O}(U^3) \,, \tag{40}
$$

with $\Delta\epsilon_{1',1} \overset{\text{def}}{=} \epsilon_{1'} - \epsilon_1$. For the correlation function in equilibrium, we get

$$
\hat{C}_q^{\text{eq.}\uparrow\uparrow} = \frac{1}{\Omega^2} \sum_k n_{k+q}(1 - n_k)
$$

$$
- \frac{2U^2}{\Omega^2} \sum_k n_{k+q}(1 - n_k) \int d\epsilon_{1'} \int d\epsilon_1 \int d\epsilon_{2'} \int d\epsilon_2 D(\epsilon_{1'})D(\epsilon_1)D(\epsilon_{2'})D(\epsilon_2)
$$

$$
\times \frac{\Delta\epsilon_{2',2}}{\Delta\epsilon_{1',1,2,2'}} \left( \frac{1}{(\Delta\epsilon_{2',2})^2 + (\Delta\epsilon_{1',1,2,2'})^2} - \frac{1}{(\Delta\epsilon_{2',2})^2} \right) (n_{1'} - n_1)(n_{2'} - n_2)
$$

$$
+ \frac{2U^2}{\Omega^2} \sum_k n_{k+q}(1 - n_k) \int d\epsilon_{1'} \int d\epsilon_1 \int d\epsilon_{2'} \int d\epsilon_2 D(\epsilon_{1'})D(\epsilon_1)D(\epsilon_{2'})D(\epsilon_2)
$$

$$
\times \frac{(\Delta\epsilon_{1',1,2,2'}) + (\Delta\epsilon_{2,2'})}{(\Delta\epsilon_{1,1'})} \left( \frac{1}{(\Delta\epsilon_{1',1,2,2'})^2 + (\Delta\epsilon_{2,2'})^2} \right) (n_{k_1'} - n_{k_1})(n_{k_2'} - n_{k_2})
$$

$$
+ \frac{U^2}{\Omega^2} \sum_k n_{k+q}(1 - n_k) \int d\epsilon_{1'} \int d\epsilon_1 \int d\epsilon_{2'} \int d\epsilon_2 D(\epsilon_{1'})D(\epsilon_1)D(\epsilon_{2'})D(\epsilon_2)
$$

$$
\times \frac{1}{(\Delta\epsilon_{1',1})(\Delta\epsilon_{2',2})}(n_{1'} - n_1)(n_{2'} - n_2)
$$

$$
+ \mathcal{O}(q^2) + \mathcal{O}(U^3) \,. \tag{41}
$$

At zero temperature, the function $\sum_k n_{k+q}(1 - n_k)$ can be geometrically estimated to be proportional to $|q|$ for small momentum. Hence, the above solutions represent the algebraically decaying parts proportional to $|x' - x|^{-2}$ in real space.

Now, we can calculate the time average of the nonequilibrium correlation function and find

the prethermalization value

$$\hat{C}_q^{\text{pre.}\uparrow\uparrow} \equiv \overline{\hat{C}_q^{\uparrow\uparrow}(t)}$$
$$= \hat{C}_q^{\text{eq.}\uparrow\uparrow} + \mathcal{O}(q^2) + \mathcal{O}(U^3) . \tag{42}$$

## 3.4 Relation to prethermalization

In generic non-integrable systems – like the Hubbard model in higher dimensions $(d > 1)$ – we expect thermalization after the system has been brought out of equilibrium [3,6]. However, a number of systems with a two-stage course of thermalization have been found, where an intermediate prethermalization regime can be identified before the entire system reaches thermal equilibrium [8–10,40]. Berges, Borsányi and Wetterich pointed out that it will be sufficient to look at the prethermalization regime if the quantity of interest (for example, the equation of state in hydrodynamical considerations) has already obtained an equilibrium value [8]. Considering heavy-ion collisions, they argued that mode quantities like the momentum distribution function memorize the initial conditions in the prethermalization regime and only decay in the long-time limit when thermalization commences to their thermal values. This leads to the formation of characteristic plateaus for momentum-dependent quantities. In contrast, local quantities that are not explicitly momentum-dependent quickly lose information on the initial state and already equilibrate in the prethermalization regime.

In the context of condensed matter systems, Moeckel and Kehrein calculated the momentum distribution function $N_k(t) = n_k + \Delta N_k(t)$ for the Hubbard quench setup considered in this paper and found that it reaches a prethermalization value

$$\Delta N_k^{\text{pre.}} = 2\Delta N_k^{\text{eq.}} + \mathcal{O}(U^3) \tag{43}$$

after the interaction quench [9]. They assumed zero temperature for the pre-quench state, while the equilibrium value $\Delta N_k^{\text{eq.}}$ was defined with respect to the interacting model at zero temperature. The leading order contribution of the prethermalization value $\Delta N_k^{\text{pre.}}$ differs from the equilibrium function at zero temperature by a factor 2. In contrast, the interaction energy, which is a sum of local terms, already reaches the equilibrium value (associated with zero temperature) in the prethermalized regime, as Moeckel and Kehrein concluded from the Feynman-Hellman theorem. Eckstein, Kollar and Werner confirmed the result from eq. (43) by numerical calculations in dynamical mean-field theory (DMFT) [10].

Another route in understanding prethermalization behavior is picturing it as near-integrability induced bottlenecks in the thermalization dynamics, as done by Kollar, Wolf and Eckstein [11]. They emphasized that thermalization in nearly integrable systems can be massively delayed the closer the system comes to integrability by the formation of prethermalization plateaus. While integrable systems usually relax to a nonthermal state that can be described by a generalized Gibbs ensemble (GGE) [7], Kollar, Wolf and Eckstein considered an interaction quench in a system with a weakly perturbed Hamiltonian $\boldsymbol{H} = \boldsymbol{H}_0 + g\boldsymbol{H}_{\text{int}}$ (starting from the integrable point $\boldsymbol{H}_0$) and showed that for conserved quantities of $\boldsymbol{H}_0$ the prethermalization value can also be predicted by a GGE. The approximate constants of motion in the perturbed system defer thermalization to time scales $t \gg \mathcal{O}(g^{-2})$ [9,11].

We now turn our attention towards the prethermalized values of nonequilibrium correlation functions calculated in this paper.

We emphasize that the pre-quench state is a thermal state of the noninteracting Hamiltonian at temperature $T$. The heating effect of the quench will increase the temperature, but only on a time scale much longer than the time scale covered in our calculation. The prethermalization regime corresponds to times before these heating effects set in, that is $t \lesssim \rho_{\mathrm{F}}^{-1} U^{-2}$. The equilibrium values $C^{\mathrm{eq.}}$ are defined with respect to equilibrium states of the interacting Hamiltonian at temperature $T$, which is equal to the temperature of the pre-quench state.

The density-density correlation function for antiparallel spins reaches a prethermalization value given by eq. (35),

$$C_{x',x}^{\mathrm{pre.}\uparrow\downarrow} = C_{x',x}^{\mathrm{eq.}\uparrow\downarrow} + \mathcal{O}(U^2) \ , \tag{44}$$

which, in leading order, is the equilibrium value of the interacting model.

We cannot find such a relation for the prethermalization value of the parallel-spin correlation function from eq. (36). But we can at least make a statement for the long-range part, which is associated with the small momentum expansion of its Fourier transform. The linear order expansion for small momentum transfer from eq. (42) yields the relation

$$\hat{C}_q^{\mathrm{pre.}\uparrow\uparrow} = \hat{C}_q^{\mathrm{eq.}\uparrow\uparrow} + \mathcal{O}(q^2) + \mathcal{O}(U^3) \ . \tag{45}$$

Hence, the long-range correlations between parallel spins show prethermal behavior equal to the equilibrium behavior.

We conclude that our findings are close to the original picture of prethermalization by Berges, Borsányi and Wetterich. While in the prethermalization regime momentum-dependent quantities like the distribution from eq. (43) differ from equilibrium, local quantities like the interaction energy or the density-density correlation functions from eqs. (44) and (45) already prethermalize to equilibrium values that are associated with the interacting model at a temperature equal to the pre-quench temperature. In the long-time behavior where the system thermalizes and that is not covered by our approach, we expect the heating effect of the quench to further increase the temperature.

## 4    Conclusion

In this paper, we calculated the time evolution of the annihilation operator in the Fermi-Hubbard model in $d > 1$ dimensions in a perturbative manner for weak interaction $U$. In particular, no secular terms appear, so that the perturbative expansion covers time scales up to and including the prethermalization regime.

We used our result to construct equal-time density-density correlation functions for antiparallel and parallel spins in a leading-order expansion. Here, we could write down the functions for both the nonequilibrium case – generated by a quench starting from the eigenstate of the noninteracting model at temperature $T$ to a weak interaction – and the equilibrium case defined by the weakly interacting model at the same temperature $T$.

For correlations between antiparallel spins, we calculated the time average of the post-quench scenario, which corresponds to the prethermalization value. We demonstrated that the prethermalization value equals the equilibrium value at temperature $T$ and explained this

result by the notion that local quantities already equilibrate in the prethermalization regime. For correlations between parallel spins, we also gave closed expressions for nonequilibrium and equilibrium, where the leading-order contributions were of second order in $U$. For a direct comparison, we needed the further approximation of an infinite dimensional lattice, where we could show that at least the long-range part of the correlation function also reaches the equilibrium value in the prethermalization regime.

Our approach is valid on time scales up to and including $t \lesssim \rho_{\mathrm{F}}^{-1} U^{-2}$ and does not include the heating effect of the quench, which would further increase the temperature and only shows on a much longer time scale where thermalization takes place.

We point out that our solutions from eqs. (25) - (27) can be used for a second order expansion of expectation values of any observable that is composed of at most four annihilation and creation operators. Therefore, one can use the perturbative expansion also for other purposes.

## Acknowledgements

We are grateful for discussions with M. Kastner and L. Cevolani.

**Funding information**   This work was supported through SFB 1073 (project B03) of the Deutsche Forschungsgemeinschaft (DFG).

M.K. is financially supported by the Bischöfliche Studienförderung Cusanuswerk.

## A   Consistency checks

### A.1   Preservation of the canonical anti-commutation relation

The application of the forward-backward scheme depicted in Fig. 1 is a sequence of unitary transformations on the annihilation and creation operators. Hence, we expect the canonical anticommutation relation

$$\left[ \boldsymbol{c}_{k\uparrow}(t), \boldsymbol{c}_{k'\uparrow}^{\dagger}(t) \right]_{+} \overset{!}{=} \delta_{k,k'} + \mathcal{O}(U^3) \tag{46}$$

to be preserved, at least up to second order in $U$. This motivates a consistency check for the time-evolved solutions from eqs. (25) - (27) after the unitary perturbation theory scheme has

been applied to the annihilation operator. With its general form from eq. (15), we get

$$
\begin{aligned}
\left[\boldsymbol{c}_{k\uparrow}(t), \boldsymbol{c}_{k'\uparrow}^{\dagger}(t)\right]_{+} =\ & h_k(t)h_{k'}^*(t)\delta_{k,k'} \\
& + \sum_{k_1,k_2',k_2} F_{k,k_1,k_2',k_2}(t)F_{k',k_1,k_2',k_2}^*(t)\big((1-n_{k_1})n_{k_2'}(1-n_{k_2}) + n_{k_1}(1-n_{k_2'})n_{k_2}\big) \\
& \quad \times \delta_{k+k_2',k_1+k_2}\delta_{k,k'} \\
& - \sum_{k_i',k_i} F_{k,k_1,k_2',k_2}(t)F_{k',k_1',k_2',k_2}^*(t)(n_{k_2'}-n_{k_2}) : \boldsymbol{c}_{k_1'\uparrow}^{\dagger}\boldsymbol{c}_{k_1\uparrow} : \delta_{k+k_2',k_1+k_2}\delta_{k'+k_1,k+k_1'} \\
& + \sum_{k_1',k_1} h_k(t)\big(G_{k',k,k_1,k_1'}^*(t) - G_{k',k_1',k_1,k}^*(t)\big) : \boldsymbol{c}_{k_1'\uparrow}^{\dagger}\boldsymbol{c}_{k_1\uparrow} : \delta_{k'+k_1,k+k_1'} \\
& + \sum_{k_1',k_1} h_{k'}^*(t)\big(G_{k,k',k_1',k_1}(t) - G_{k,k_1,k_1',k'}(t)\big) : \boldsymbol{c}_{k_1'\uparrow}^{\dagger}\boldsymbol{c}_{k_1\uparrow} : \delta_{k'+k_1,k+k_1'} \\
& + \text{linearly independent terms} \\
& + \mathcal{O}(U^3)\,,
\end{aligned}
\tag{47}
$$

where we only consider terms that have an operator structure proportional to $\mathbf{1}$ or $: \boldsymbol{c}_{k_1'\uparrow}^{\dagger}\boldsymbol{c}_{k_1\uparrow} :$. Thus, we have two consistency conditions,

$$
\begin{aligned}
1 \overset{!}{=}\ & |h_k(t)|^2 \\
& + \sum_{k_1,k_2',k_2} |F_{k,k_1,k_2',k_2}(t)|^2\big((1-n_{k_1})n_{k_2'}(1-n_{k_2}) + n_{k_1}(1-n_{k_2'})n_{k_2}\big)\delta_{k+k_2',k_1+k_2} \\
& + \mathcal{O}(U^3)\,, \\[4pt]
0 \overset{!}{=}\ & -\sum_{k_2',k_2} |F_{k,k_1,k_2',k_2}(t)|^2(n_{k_2'}-n_{k_2})\delta_{k+k_2',k_1+k_2} \\
& + h_k(t)\big(G_{k,k,k_1,k_1}^*(t) - G_{k,k_1,k_1,k}^*(t)\big) \\
& + h_k^*(t)\big(G_{k,k,k_1,k_1}(t) - G_{k,k_1,k_1,k}(t)\big) \\
& + \mathcal{O}(U^3)\,,
\end{aligned}
$$

with equation numbers (48) and (49) respectively.

which are two relations that our three solutions from eqs. (25) - (27) should fulfill. We insert the solutions into the first relation and get

$$1 \overset{!}{=} 1$$

$$- \frac{U^2}{\Omega^2} \sum_{k_1,k_2',k_2} \frac{1 - e^{-i(\Delta\epsilon_{k,k_1,k_2',k_2})t}}{(\Delta\epsilon_{k,k_1,k_2',k_2})^2}$$

$$\times ((1 - n_{k_1})n_{k_2'}(1 - n_{k_2}) + n_{k_1}(1 - n_{k_2'})n_{k_2})\delta_{k+k_2',k_1+k_2}$$

$$- \frac{U^2}{\Omega^2} \sum_{k_1,k_2',k_2} \frac{1 - e^{i(\Delta\epsilon_{k,k_1,k_2',k_2})t}}{(\Delta\epsilon_{k,k_1,k_2',k_2})^2}$$

$$\times ((1 - n_{k_1})n_{k_2'}(1 - n_{k_2}) + n_{k_1}(1 - n_{k_2'})n_{k_2})\delta_{k+k_2',k_1+k_2}$$

$$+ \frac{U^2}{\Omega^2} \sum_{k_1,k_2',k_2} \frac{\left|1 - e^{i(\Delta\epsilon_{k,k_1,k_2',k_2})t}\right|^2}{(\Delta\epsilon_{k,k_1,k_2',k_2})^2} ((1 - n_{k_1})n_{k_2'}(1 - n_{k_2}) + n_{k_1}(1 - n_{k_2'})n_{k_2})\delta_{k+k_2',k_1+k_2}$$

$$+ \mathcal{O}(U^3) \,. \tag{50}$$

We recognize that the last three terms cancel out. Thus, the first consistency condition is fulfilled.
The second relation requires

$$0 \overset{!}{=} - \frac{U^2}{\Omega^2} \sum_{k_2',k_2} \frac{\left|1 - e^{i(\Delta\epsilon_{k,k_1,k_2',k_2})t}\right|^2}{(\Delta\epsilon_{k,k_1,k_2',k_2})^2} (n_{k_2'} - n_{k_2})\delta_{k+k_2',k_1+k_2}$$

$$- G^*_{k,k_1,k_1,k}(t)$$

$$- G_{k,k_1,k_1,k}(t)$$

$$+ \mathcal{O}(U^3) \,, \tag{51}$$

where we have used that $G_{k,k,k_1,k_1}(t) = 0 + \mathcal{O}(U^3)$. Furthermore, eq. (27) implies

$$G_{k,k_1,k_1,k}(t) = - \frac{U^2}{\Omega^2} \sum_{k_3',k_3} \frac{\Delta\epsilon_{k_3',k_3,k_1,k}}{\Delta\epsilon_{k,k_1,k_3,k_3'}} \left( \frac{1 - e^{i(\Delta\epsilon_{k,k_1,k_1,k})t}}{(\Delta\epsilon_{k_3',k_3,k_1,k})^2 + (\Delta\epsilon_{k,k_1,k_3,k_3'})^2} \right.$$

$$\left. - \frac{e^{i(\Delta\epsilon_{k,k_1,k_3,k_3'})t} - e^{i(\Delta\epsilon_{k,k_1,k_1,k})t}}{(\Delta\epsilon_{k_3',k_3,k_1,k})^2} \right)$$

$$\times (n_{k_3'} - n_{k_3})\delta_{k_1+k_3',k+k_3}$$

$$+ \mathcal{O}(U^3)$$

$$= - \frac{U^2}{\Omega^2} \sum_{k_2',k_2} \frac{1 - e^{i(\Delta\epsilon_{k,k_1,k_2',k_2})t}}{(\Delta\epsilon_{k,k_1,k_2',k_2})^2} (n_{k_2'} - n_{k_2})\delta_{k+k_2',k_1+k_2}$$

$$+ \mathcal{O}(U^3) \,. \tag{52}$$

With this result, the second relation reads

$$
\begin{aligned}
0 \overset{!}{=} \; & -\frac{U^2}{\Omega^2} \sum_{k_2',k_2} \frac{\left|1 - e^{i(\Delta\epsilon_{k,k_1,k_2',k_2})t}\right|^2}{(\Delta\epsilon_{k,k_1,k_2',k_2})^2} (n_{k_2'} - n_{k_2})\delta_{k+k_2',k_1+k_2} \\
& + \frac{U^2}{\Omega^2} \sum_{k_2',k_2} \frac{1 - e^{-i(\Delta\epsilon_{k,k_1,k_2',k_2})t}}{(\Delta\epsilon_{k,k_1,k_2',k_2})^2} (n_{k_2'} - n_{k_2})\delta_{k+k_2',k_1+k_2} \\
& + \frac{U^2}{\Omega^2} \sum_{k_2',k_2} \frac{1 - e^{i(\Delta\epsilon_{k,k_1,k_2',k_2})t}}{(\Delta\epsilon_{k,k_1,k_2',k_2})^2} (n_{k_2'} - n_{k_2})\delta_{k+k_2',k_1+k_2} \\
& + \mathcal{O}(U^3)\,,
\end{aligned}
\tag{53}
$$

which is clearly fulfilled due to cancelation of all terms on the right-hand side.

## A.2   Total spin-up particle number

The total spin-up particle number

$$
\boldsymbol{N}_\uparrow \overset{\text{def}}{=} \sum_k \boldsymbol{c}_{k\uparrow}^\dagger \boldsymbol{c}_{k\uparrow}
\tag{54}
$$

is a conserved quantity in the Hubbard model, because it commutes with the Hamiltonian. This provides another consistency check for the solutions from eqs. (25) - (27). From eq. (63), we can directly construct the time-evolved total spin-up particle number operator, where we only focus on terms with an operator structure proportional to $\boldsymbol{1}$ or $: \boldsymbol{c}_{k_1'\uparrow}^\dagger \boldsymbol{c}_{k_1\uparrow} :$,

$$
\begin{aligned}
\boldsymbol{N}_\uparrow(t) = & \sum_k |h_k(t)|^2 n_k + \sum_{k,k_1,k_2',k_2} |F_{k,k_1,k_2',k_2}(t)|^2 n_{k_1}(1 - n_{k_2'})n_{k_2}\delta_{k+k_2',k_1+k_2} \\
& + \sum_{k_1} |h_{k_1}(t)|^2 : \boldsymbol{c}_{k_1\uparrow}^\dagger \boldsymbol{c}_{k_1\uparrow} : \\
& + \sum_{k_1',k_1,k_2',k_2} |F_{k_1',k_1,k_2',k_2}(t)|^2(1 - n_{k_2'})n_{k_2} : \boldsymbol{c}_{k_1\uparrow}^\dagger \boldsymbol{c}_{k_1\uparrow} : \delta_{k_1'+k_2',k_1+k_2} \\
& + \sum_{k_1',k_1} h_{k_1'}^*(t)\big(G_{k_1',k_1',k_1,k_1}(t) - G_{k_1',k_1,k_1,k_1'}(t)\big)n_{k_1'} : \boldsymbol{c}_{k_1\uparrow}^\dagger \boldsymbol{c}_{k_1\uparrow} : \\
& + \sum_{k_1',k_1} h_{k_1'}(t)\big(G_{k_1',k_1',k_1,k_1}^*(t) - G_{k_1',k_1,k_1,k_1'}^*(t)\big)n_{k_1'} : \boldsymbol{c}_{k_1\uparrow}^\dagger \boldsymbol{c}_{k_1\uparrow} : \\
& + \text{linearly independent terms} \\
& + \mathcal{O}(U^3)\,.
\end{aligned}
\tag{55}
$$

We insert the expressions from eqs. (25) - (27) and get

$$
\begin{aligned}
\boldsymbol{N}_\uparrow(t) = \sum_k n_k &- \frac{2U^2}{\Omega^2} \sum_{k,k_1,k_2',k_2} \frac{1 - \cos\left((\Delta\epsilon_{k,k_1,k_2',k_2})t\right)}{(\Delta\epsilon_{k,k_1,k_2',k_2})^2} \\
&\times n_k\left((1-n_{k_1})n_{k_2'}(1-n_{k_2}) + n_{k_1}(1-n_{k_2'})n_{k_2}\right)\delta_{k+k_2',k_1+k_2} \\
+ \frac{2U^2}{\Omega^2} &\sum_{k,k_1,k_2',k_2} \frac{1 - \cos\left((\Delta\epsilon_{k,k_1,k_2',k_2})t\right)}{(\Delta\epsilon_{k,k_1,k_2',k_2})^2} n_{k_1}(1-n_{k_2'})n_{k_2}\delta_{k+k_2',k_1+k_2} \\
+ \sum_{k_1} &: \boldsymbol{c}^\dagger_{k_1\uparrow}\boldsymbol{c}_{k_1\uparrow} : \\
- \frac{2U^2}{\Omega^2} &\sum_{k_1',k_1,k_2',k_2} \frac{1 - \cos\left((\Delta\epsilon_{k_1',k_1,k_2',k_2})t\right)}{(\Delta\epsilon_{k_1',k_1,k_2',k_2})^2} \\
&\times \left((1-n_{k_1'})n_{k_2}(1-n_{k_2'}) + n_{k_1'}(1-n_{k_2})n_{k_2'}\right) : \boldsymbol{c}^\dagger_{k_1\uparrow}\boldsymbol{c}_{k_1\uparrow} : \delta_{k_1'+k_2',k_1+k_2} \\
+ \frac{2U^2}{\Omega^2} &\sum_{k_1',k_1,k_2',k_2} \frac{1 - \cos\left((\Delta\epsilon_{k_1',k_1,k_2',k_2})t\right)}{(\Delta\epsilon_{k_1',k_1,k_2',k_2})^2}(1-n_{k_2'})n_{k_2} : \boldsymbol{c}^\dagger_{k_1\uparrow}\boldsymbol{c}_{k_1\uparrow} : \delta_{k_1'+k_2',k_1+k_2} \\
+ \frac{2U^2}{\Omega^2} &\sum_{k_1',k_1,k_2',k_2} \frac{1 - \cos\left((\Delta\epsilon_{k_1',k_1,k_2',k_2})t\right)}{(\Delta\epsilon_{k_1',k_1,k_2',k_2})^2} \\
&\times n_{k_1'}\left(n_{k_2'}(1-n_{k_2}) - (1-n_{k_2'})n_{k_2}\right) : \boldsymbol{c}^\dagger_{k_1\uparrow}\boldsymbol{c}_{k_1\uparrow} : \delta_{k_1'+k_2',k_1+k_2} \\
&+ \text{linearly independent terms} \\
&+ \mathcal{O}(U^3) \,.
\end{aligned}
\tag{56}
$$

We convince ourselves that after interchanging indices most of the terms cancel out, yielding

$$
\boldsymbol{N}_\uparrow(t) = \sum_k n_k + \sum_k : \boldsymbol{c}^\dagger_{k\uparrow}\boldsymbol{c}_{k\uparrow} : + \text{linearly independent terms} + \mathcal{O}(U^3) \,.
\tag{57}
$$

As this is time-independent, this part of the operator is consistent with the conservation of the total spin-up particle number.

### A.3 Variance of the total spin-up particle number

As the total spin-up particle number $\boldsymbol{N}_\uparrow$ is a conserved quantity, also its variance

$$
\left\langle \boldsymbol{N}_\uparrow^2 \right\rangle - \left\langle \boldsymbol{N}_\uparrow \right\rangle^2
\tag{58}
$$

should be time-independent. We can construct the variance from the equal-time connected density-density correlation function for parallel spins, $C_{x',x}^{\uparrow\uparrow}(t)$, by a summation over $x'$ and $x$,

$$
\begin{aligned}
\sum_{x',x} C_{x',x}^{\uparrow\uparrow}(t) &= \sum_{x',x}\langle \boldsymbol{n}_{x'\uparrow}(t)\boldsymbol{n}_{x\uparrow}(t)\rangle - \sum_{x',x}\langle \boldsymbol{n}_{x'\uparrow}(t)\rangle\langle \boldsymbol{n}_{x\uparrow}(t)\rangle \\
&= \left\langle \boldsymbol{N}_\uparrow^2 \right\rangle - \left\langle \boldsymbol{N}_\uparrow \right\rangle^2 \,.
\end{aligned}
\tag{59}
$$

The solution for $C^{\uparrow\uparrow}_{x',x}(t)$ from eq. (36) should be consistent with this. The summation over $x'$ yields a $\delta_{k',k}$ and we get

$$
\begin{aligned}
\sum_{x',x} C^{\uparrow\uparrow}_{x',x}(t) = &\sum_k n_k(1-n_k) \\
&- \frac{4U^2}{\Omega^2} \sum_k n_k(1-n_k) \sum_{k_1,k'_2,k_2} \frac{1-\cos\left((\Delta\epsilon_{k,k_1,k'_2,k_2})t\right)}{(\Delta\epsilon_{k,k_1,k'_2,k_2})^2} \\
&\qquad\qquad\qquad\times \left((1-n_{k_1})n_{k'_2}(1-n_{k_2}) + n_{k_1}(1-n_{k'_2})n_{k_2}\right)\delta_{k+k'_2,k_1+k_2} \\
&+ \frac{4U^2}{\Omega^2} \sum_k n_k(1-n_k) \sum_{k_1,k'_2,k_2} \frac{1-\cos\left((\Delta\epsilon_{k,k_1,k'_2,k_2})t\right)}{(\Delta\epsilon_{k,k_1,k'_2,k_2})^2} n_{k'_2}(1-n_{k_2})\delta_{k+k'_2,k_1+k_2} \\
&- \frac{4U^2}{\Omega^2} \sum_k n_k(1-n_k) \sum_{k_1,k'_2,k_2} \frac{1-\cos\left((\Delta\epsilon_{k,k_1,k'_2,k_2})t\right)}{(\Delta\epsilon_{k,k_1,k'_2,k_2})^2} n_{k_1}(n_{k'_2}-n_{k_2})\delta_{k+k'_2,k_1+k_2} \\
&- \frac{4U^2}{\Omega^2} \sum_k n_k \sum_{k_1,k'_2,k_2} \frac{1-\cos\left((\Delta\epsilon_{k,k_1,k'_2,k_2})t\right)}{(\Delta\epsilon_{k,k_1,k'_2,k_2})^2} (1-n_{k_1})n_{k'_2}(1-n_{k_2})\delta_{k+k'_2,k_1+k_2} \\
&+ \frac{4U^2}{\Omega^2} \sum_k n_k \sum_{k_1,k'_2,k_2} \frac{1-\cos\left((\Delta\epsilon_{k,k_1,k'_2,k_2})t\right)}{(\Delta\epsilon_{k,k_1,k'_2,k_2})^2} (1-n_{k_1})n_{k'_2}(1-n_{k_2})\delta_{k+k'_2,k_1+k_2} \\
&+ \mathcal{O}(U^3) \,. \qquad\qquad\qquad\qquad\qquad\qquad\qquad\qquad\qquad\qquad (60)
\end{aligned}
$$

The last two terms cancel out directly and the other terms can be rearranged such that

$$
\begin{aligned}
\sum_{x',x} C^{\uparrow\uparrow}_{x',x}(t) = &\sum_k n_k(1-n_k) \\
&- \frac{4U^2}{\Omega^2} \sum_k n_k(1-n_k) \sum_{k_1,k'_2,k_2} \frac{1-\cos\left((\Delta\epsilon_{k,k_1,k'_2,k_2})t\right)}{(\Delta\epsilon_{k,k_1,k'_2,k_2})^2} \\
&\qquad\qquad\qquad\times \left((1-n_{k_1})n_{k'_2}(1-n_{k_2}) + n_{k_1}(1-n_{k'_2})n_{k_2}\right)\delta_{k+k'_2,k_1+k_2} \\
&+ \frac{4U^2}{\Omega^2} \sum_k n_k(1-n_k) \sum_{k_1,k'_2,k_2} \frac{1-\cos\left((\Delta\epsilon_{k,k_1,k'_2,k_2})t\right)}{(\Delta\epsilon_{k,k_1,k'_2,k_2})^2} \\
&\qquad\qquad\qquad\times \left((1-n_{k_1})n_{k'_2}(1-n_{k_2}) + n_{k_1}(1-n_{k'_2})n_{k_2}\right)\delta_{k+k'_2,k_1+k_2} \\
&+ \mathcal{O}(U^3) \\
= &\sum_k n_k(1-n_k) \\
&+ \mathcal{O}(U^3) \,. \qquad\qquad\qquad\qquad\qquad\qquad\qquad\qquad\qquad\qquad (61)
\end{aligned}
$$

This is clearly time-independent and hence consistent with the conservation of the variance of the total spin-up particle number.

# B  Calculation of correlation functions

Given the general structure of the annihilation operator from eq. (15), we first calculate

$$
\begin{aligned}
c_{k'\uparrow}^{\dagger}(t)c_{k\uparrow}(t) = {}& h_{k'}^{*}(t)h_{k}(t) : c_{k'\uparrow}^{\dagger} :: c_{k\uparrow} : \\
& + \sum_{k_1,k_2',k_2} h_{k'}^{*}(t)F_{k,k_1,k_2',k_2}(t) : c_{k'\uparrow}^{\dagger} :: c_{k_1\uparrow}c_{k_2'\downarrow}^{\dagger}c_{k_2\downarrow} : \delta_{k+k_2',k_1+k_2} \\
& + \sum_{k_1,k_2',k_2} h_{k'}^{*}(t)G_{k,k_1,k_2',k_2}(t) : c_{k'\uparrow}^{\dagger} :: c_{k_1\uparrow}c_{k_2'\uparrow}^{\dagger}c_{k_2\uparrow} : \delta_{k+k_2',k_1+k_2} \\
& + \sum_{k_1,k_2',k_2} F_{k',k_1,k_2',k_2}^{*}(t)h_{k}(t) : c_{k_2\downarrow}^{\dagger}c_{k_2'\downarrow}c_{k_1\uparrow}^{\dagger} :: c_{k\uparrow} : \delta_{k'+k_2',k_1+k_2} \\
& + \sum_{k_i',k_i} F_{k',k_1,k_2',k_2}^{*}(t)F_{k,k_1',k_3',k_3}(t) : c_{k_2\downarrow}^{\dagger}c_{k_2'\downarrow}c_{k_1\uparrow}^{\dagger} :: c_{k_1'\uparrow}c_{k_3'\downarrow}^{\dagger}c_{k_3\downarrow} : \delta_{k'+k_2',k_1+k_2}\delta_{k+k_3',k_1'+k_3} \\
& + \sum_{k_1,k_2',k_2} G_{k',k_1,k_2',k_2}^{*}(t)h_{k}(t) : c_{k_2\uparrow}^{\dagger}c_{k_2'\uparrow}c_{k_1\uparrow}^{\dagger} :: c_{k\uparrow} : \delta_{k'+k_2',k_1+k_2} \\
& + \text{"irrelevant terms"} \\
& + \mathcal{O}(U^3) \,,
\end{aligned}
\tag{62}
$$

where normal-ordered products of at least four annihilation and creation operators that are of second order in $U$ are shifted into the irrelevant terms.

The next step is calculating the products of normal-ordered expressions, which yields

$$
\begin{aligned}
\boldsymbol{c}_{k'\uparrow}^{\dagger}(t)\boldsymbol{c}_{k\uparrow}(t) = {}& |h_{k'}(t)|^2 n_{k'}\delta_{k',k} + \sum_{k_1,k_2',k_2} |F_{k',k_1,k_2',k_2}(t)|^2 n_{k_1}(1-n_{k_2'})n_{k_2}\delta_{k'+k_2',k_1+k_2}\delta_{k',k} \\
& + h_{k'}^{*}(t)h_k(t):\boldsymbol{c}_{k'\uparrow}^{\dagger}\boldsymbol{c}_{k\uparrow}: \\
& + \sum_{k_i',k_i} F_{k',k_1',k_2',k_2}^{*}(t)F_{k,k_1,k_2',k_2}(t)(1-n_{k_2'})n_{k_2}:\boldsymbol{c}_{k_1'\uparrow}^{\dagger}\boldsymbol{c}_{k_1\uparrow}:\delta_{k+k_2',k_1+k_2}\delta_{k'+k_1,k+k_1'} \\
& + \sum_{k_1',k_1} h_{k'}^{*}(t)\big(G_{k,k',k_1',k_1}(t) - G_{k,k_1,k_1',k'}(t)\big)n_{k'}:\boldsymbol{c}_{k_1'\uparrow}^{\dagger}\boldsymbol{c}_{k_1\uparrow}:\delta_{k'+k_1,k+k_1'} \\
& + \sum_{k_1',k_1} h_k(t)\big(G_{k',k,k_1,k_1'}^{*}(t) - G_{k',k_1',k_1,k}^{*}(t)\big)n_k:\boldsymbol{c}_{k_1'\uparrow}^{\dagger}\boldsymbol{c}_{k_1\uparrow}:\delta_{k'+k_1,k+k_1'} \\
& + \sum_{k_1',k_1} h_{k'}^{*}(t)F_{k,k',k_1',k_1}(t)n_{k'}:\boldsymbol{c}_{k_1'\downarrow}^{\dagger}\boldsymbol{c}_{k_1\downarrow}:\delta_{k'+k_1,k+k_1'} \\
& + \sum_{k_1',k_1} h_k(t)F_{k',k,k_1,k_1'}^{*}(t)n_k:\boldsymbol{c}_{k_1'\downarrow}^{\dagger}\boldsymbol{c}_{k_1\downarrow}:\delta_{k'+k_1,k+k_1'} \\
& + \sum_{k_i',k_i} F_{k',k_2,k_2',k_1'}^{*}(t)F_{k,k_2,k_2',k_1}(t)(1-n_{k_2'})n_{k_2}:\boldsymbol{c}_{k_1'\downarrow}^{\dagger}\boldsymbol{c}_{k_1\downarrow}:\delta_{k+k_2',k_1+k_2}\delta_{k'+k_1,k+k_1'} \\
& - \sum_{k_i',k_i} F_{k',k_2',k_1,k_2}^{*}(t)F_{k,k_2',k_1',k_2}(t)n_{k_2'}n_{k_2}:\boldsymbol{c}_{k_1'\downarrow}^{\dagger}\boldsymbol{c}_{k_1\downarrow}:\delta_{k'+k_1,k_2'+k_2}\delta_{k'+k_1,k+k_1'} \\
& + \sum_{k_1,k_2',k_2} h_{k'}^{*}(t)F_{k,k_1,k_2',k_2}(t):\boldsymbol{c}_{k'\uparrow}^{\dagger}\boldsymbol{c}_{k_1\uparrow}\boldsymbol{c}_{k_2'\downarrow}^{\dagger}\boldsymbol{c}_{k_2\downarrow}:\delta_{k+k_2',k_1+k_2} \\
& + \sum_{k_1',k_2',k_2} h_k(t)F_{k',k_1',k_2,k_2'}^{*}(t):\boldsymbol{c}_{k_1'\uparrow}^{\dagger}\boldsymbol{c}_{k\uparrow}\boldsymbol{c}_{k_2'\downarrow}^{\dagger}\boldsymbol{c}_{k_2\downarrow}:\delta_{k_1'+k_2',k'+k_2} \\
& + \text{"irrelevant terms"} \\
& + \mathcal{O}(U^3) \,.
\end{aligned}
\tag{63}
$$

## B.1  Antiparallel-spin correlations

For anti-parallel spins, the equal-time correlation function has a first-order contribution in $U$. Therefore, we neglect the second-order terms in eq. (63). The last two terms only completely contract among each other, which would be of second order, hence they are irrelevant. We only need

$$
\begin{aligned}
\boldsymbol{c}_{k'\uparrow}^{\dagger}(t)\boldsymbol{c}_{k\uparrow}(t) = {}& h_{k'}^{*}(t)h_k(t):\boldsymbol{c}_{k'\uparrow}^{\dagger}\boldsymbol{c}_{k\uparrow}: \\
& + \sum_{k_1',k_1} h_{k'}^{*}(t)F_{k,k',k_1',k_1}(t)n_{k'}:\boldsymbol{c}_{k_1'\downarrow}^{\dagger}\boldsymbol{c}_{k_1\downarrow}:\delta_{k'+k_1,k+k_1'} \\
& + \sum_{k_1',k_1} h_k(t)F_{k',k,k_1,k_1'}^{*}(t)n_k:\boldsymbol{c}_{k_1'\downarrow}^{\dagger}\boldsymbol{c}_{k_1\downarrow}:\delta_{k'+k_1,k+k_1'} \\
& + \text{"irrelevant terms"} \\
& + \mathcal{O}(U^2) \,.
\end{aligned}
\tag{64}
$$

We calcuclate the contractions between the above equation and its spin-down counterpart, yielding

$$
\begin{aligned}
C_{x',x}^{\uparrow\downarrow}(t) = & \frac{1}{\Omega^2} \sum_{k',k,q',q} e^{i(k'-k)(x'-x)} h_{k'}^*(t) h_k(t) h_{q'}^*(t) F_{q,q',k,k'}(t) n_{k'}(1-n_k) n_{q'} \delta_{k'+q',k+q} \\
& + \frac{1}{\Omega^2} \sum_{k',k,q',q} e^{i(k'-k)(x'-x)} h_{k'}^*(t) h_k(t) h_q(t) F_{q',q,k',k}^*(t) n_{k'}(1-n_k) n_q \delta_{k'+q',k+q} \\
& + \frac{1}{\Omega^2} \sum_{k',k,q',q} e^{i(k'-k)(x'-x)} h_{k'}(t) h_k^*(t) h_q^*(t) F_{q',q,k',k}(t) n_{k'}(1-n_k) n_q \delta_{k'+q',k+q} \\
& + \frac{1}{\Omega^2} \sum_{k',k,q',q} e^{i(k'-k)(x'-x)} h_{k'}(t) h_k^*(t) h_{q'}(t) F_{q,q',k,k'}^*(t) n_{k'}(1-n_k) n_{q'} \delta_{k'+q',k+q} \\
& + \mathcal{O}(U^2) \,.
\end{aligned}
\tag{65}
$$

When we insert the coefficients from eqs. (25) and (26), we arrive at the nonequilibrium solution in eq. (32), while the coefficients from eqs. (19) and (20) for $B = \infty$ give the equilibrium solution in eq. (33).

## B.2 Parallel-spin correlations

For parallel spins, the equal-time correlation function is of second order in $U$ and hence all terms in eq. (63) are relevant. Calculating all full contractions of normal-ordered expressions, we find that

$$
\begin{aligned}
C_{x',x}^{\uparrow\uparrow}(t) = & \frac{1}{\Omega^2} \sum_{k',k} e^{i(k'-k)(x'-x)} n_{k'}(1-n_k) |h_{k'}(t)|^2 |h_k(t)|^2 \\
& + \frac{2}{\Omega^2} \sum_{k',k} e^{i(k'-k)(x'-x)} n_{k'}(1-n_k) \\
& \qquad \times \sum_{k_i',k_i} \Re\left( h_{k'}^*(t) h_k(t) F_{k_1',k',k_2',k_2}(t) F_{k_1,k,k_2',k_2}^*(t) \right)(1-n_{k_2'}) n_{k_2} \delta_{k_1'+k_2',k'+k_2} \delta_{k'+k_1,k+k_1'} \\
& + \frac{2}{\Omega^2} \sum_{k',k} e^{i(k'-k)(x'-x)} n_{k'}(1-n_k) \\
& \qquad \times \sum_{k_1',k_1} \Re\left( h_{k'}^*(t) h_k(t) h_{k_1'}(t) \left( G_{k_1,k_1',k',k}^*(t) - G_{k_1,k,k',k_1'}^*(t) \right) \right) n_{k_1'} \delta_{k'+k_1,k+k_1'} \\
& + \frac{2}{\Omega^2} \sum_{k',k} e^{i(k'-k)(x'-x)} n_{k'}(1-n_k) \\
& \qquad \times \sum_{k_1',k_1} \Re\left( h_{k'}^*(t) h_k(t) h_{k_1}^*(t) \left( G_{k_1',k_1,k,k'}(t) - G_{k_1',k',k,k_1}(t) \right) \right) n_{k_1} \delta_{k'+k_1,k+k_1'}
\end{aligned}
$$

$$+ \frac{1}{\Omega^2} \sum_{k',k} e^{i(k'-k)(x'-x)} n_{k'}(1 - n_k)$$

$$\times \sum_{k'_i,k_i} h^*_{k'_1}(t) F_{k_1,k'_1,k',k}(t) h^*_{k'_2}(t) F_{k_2,k'_2,k,k'}(t) n_{k'_1} n_{k'_2} \delta_{k'_1+k'_2,k_1+k_2} \delta_{k'+k_1,k+k'_1}$$

$$+ \frac{1}{\Omega^2} \sum_{k',k} e^{i(k'-k)(x'-x)} n_{k'}(1 - n_k)$$

$$\times \sum_{k'_i,k_i} h^*_{k'_1}(t) F_{k_1,k'_1,k',k}(t) h_{k_2}(t) F^*_{k'_2,k_2,k',k}(t) n_{k'_1} n_{k_2} \delta_{k'_1+k'_2,k_1+k_2} \delta_{k'+k_1,k+k'_1}$$

$$+ \frac{1}{\Omega^2} \sum_{k',k} e^{i(k'-k)(x'-x)} n_{k'}(1 - n_k)$$

$$\times \sum_{k'_i,k_i} h_{k_1}(t) F^*_{k'_1,k_1,k,k'}(t) h^*_{k'_2}(t) F_{k_2,k'_2,k,k'}(t) n_{k_1} n_{k'_2} \delta_{k'_1+k'_2,k_1+k_2} \delta_{k'+k_1,k+k'_1}$$

$$+ \frac{1}{\Omega^2} \sum_{k',k} e^{i(k'-k)(x'-x)} n_{k'}(1 - n_k)$$

$$\times \sum_{k'_i,k_i} h_{k_1}(t) F^*_{k'_1,k_1,k,k'}(t) h_{k_2}(t) F^*_{k'_2,k_2,k',k}(t) n_{k_1} n_{k_2} \delta_{k'_1+k'_2,k_1+k_2} \delta_{k'+k_1,k+k'_1}$$

$$+ \frac{2}{\Omega^2} \sum_{k',k} e^{i(k'-k)(x'-x)} n_{k'}$$

$$\times \sum_{k'_i,k_i} \Re\big(h^*_{k'}(t) h^*_{k_1}(t) F_{k'_1,k',k_2,k'_2}(t) F_{k,k_1,k'_2,k_2}(t)\big)(1 - n_{k_1}) n_{k'_2}(1 - n_{k_2}) \delta_{k+k'_2,k_1+k_2} \delta_{k'+k_1,k+k'_1}$$

$$+ \frac{1}{\Omega^2} \sum_{k',k} e^{i(k'-k)(x'-x)} \sum_{k_1,k'_2,k_2} |h_{k'}(t)|^2 |F_{k,k_1,k'_2,k_2}(t)|^2 n_{k'}(1 - n_{k_1}) n_{k'_2}(1 - n_{k_2}) \delta_{k+k'_2,k_1+k_2}$$

$$+ \frac{1}{\Omega^2} \sum_{k',k} e^{-i(k'-k)(x'-x)} \sum_{k_1,k'_2,k_2} |h_{k'}(t)|^2 |F_{k,k_1,k'_2,k_2}(t)|^2 (1 - n_{k'}) n_{k_1}(1 - n_{k'_2}) n_{k_2} \delta_{k+k'_2,k_1+k_2}$$

$$+ \mathcal{O}(U^3) . \tag{66}$$

Again, inserting the coefficients from eqs. (25) - (27) will give us the nonequilibrium solution, while the coefficients from eqs. (19) - (21) for $B = \infty$ give the equilibrium solution.

## B.3   Limit of infinite spatial dimensions

The Fourier transformation to momentum space, $\hat{C}_q^{\uparrow\uparrow}(t) \stackrel{\text{def}}{=} \Omega^{-1} \sum_{x'-x} e^{-iq(x'-x)} C_{x',x}^{\uparrow\uparrow}(t)$, effectively yields a factor $\delta_{q,k'-k}$, and we get

$$
\hat{C}_q^{\uparrow\uparrow}(t) = \frac{1}{\Omega^2} \sum_k n_{k+q}(1-n_k)
$$

$$
- \frac{4U^2}{\Omega^4} \sum_k n_{k+q}(1-n_k)
$$

$$
\times \sum_{k_1,k_2',k_2} \frac{1 - \cos\left((\Delta\epsilon_{k,k_1,k_2',k_2})t\right)}{(\Delta\epsilon_{k,k_1,k_2',k_2})^2}
$$

$$
\times \left((1-n_{k_1})n_{k_2'}(1-n_{k_2}) + n_{k_1}(1-n_{k_2'})n_{k_2}\right)\delta_{k+k_2',k_1+k_2}
$$

$$
+ \frac{2U^2}{\Omega^4} \sum_k n_{k+q}(1-n_k)
$$

$$
\times \sum_{k_i',k_i} \frac{1 - \cos\left((\Delta\epsilon_{k+q,k_1',k_2',k_2})t\right) - \cos\left((\Delta\epsilon_{k,k_1,k_2',k_2})t\right) + \cos\left((\Delta\epsilon_{k+q,k,k_1,k_1'})t\right)}{(\Delta\epsilon_{k+q,k_1',k_2',k_2})(\Delta\epsilon_{k,k_1,k_2',k_2})}
$$

$$
\times n_{k_2'}(1-n_{k_2})\delta_{k+k_2',k_1+k_2}\delta_{k_1'-k_1,q}
$$

$$
- \frac{2U^2}{\Omega^4} \sum_k n_{k+q}(1-n_k)
$$

$$
\times \sum_{k_i',k_i} \frac{\Delta\epsilon_{k+q,k,k_2',k_2}}{\Delta\epsilon_{k_1',k_1,k_2,k_2'}} \left( \frac{1 - \cos\left((\Delta\epsilon_{k+q,k,k_1',k_1})t\right)}{(\Delta\epsilon_{k+q,k,k_2',k_2})^2 + (\Delta\epsilon_{k_1',k_1,k_2,k_2'})^2} - \frac{1 - \cos\left((\Delta\epsilon_{k+q,k,k_2',k_2})t\right)}{(\Delta\epsilon_{k+q,k,k_2',k_2})^2} \right)
$$

$$
\times (n_{k_1'} - n_{k_1})(n_{k_2'} - n_{k_2})\delta_{k_1'-k_1,q}\delta_{k_2'-k_2,q}
$$

$$
+ \frac{2U^2}{\Omega^4} \sum_k \left(n_{k+q}(1-n_k) + (1-n_{k+q})n_k\right)
$$

$$
\times \sum_{k_i',k_i} \frac{\Delta\epsilon_{k,k_1,k_2',k_2}}{\Delta\epsilon_{k+q,k_1',k_2',k_2}} \left( \frac{1 - \cos\left((\Delta\epsilon_{k+q,k,k_1,k_1'})t\right)}{(\Delta\epsilon_{k+q,k_1',k_2',k_2})^2 + (\Delta\epsilon_{k,k_1,k_2',k_2})^2} - \frac{1 - \cos\left((\Delta\epsilon_{k,k_1,k_2',k_2})t\right)}{(\Delta\epsilon_{k,k_1,k_2',k_2})^2} \right)
$$

$$
\times n_{k_1}(n_{k_2'} - n_{k_2})\delta_{k+k_2',k_1+k_2}\delta_{k_1'-k_1,q}
$$

$$
+ \frac{2U^2}{\Omega^4} \sum_k n_{k+q}(1-n_k)
$$

$$
\times \sum_{k_i',k_i} \frac{(\Delta\epsilon_{k_1',k_1,k_2,k_2'}) + (\Delta\epsilon_{k+q,k,k_2,k_2'})}{(\Delta\epsilon_{k+q,k,k_1,k_1'})} \left( \frac{1 - \cos\left((\Delta\epsilon_{k+q,k,k_1,k_1'})t\right)}{(\Delta\epsilon_{k_1',k_1,k_2,k_2'})^2 + (\Delta\epsilon_{k+q,k,k_2,k_2'})^2} \right)
$$

$$
\times (n_{k_1'} - n_{k_1})(n_{k_2'} - n_{k_2})\delta_{k_1'-k_1,q}\delta_{k_2'-k_2,q}
$$

$$
+ \frac{2U^2}{\Omega^4} \sum_k \left( n_{k+q}(1 - n_k) + (1 - n_{k+q})n_k \right)
$$

$$
\times \sum_{k_i', k_i} \frac{(\Delta\epsilon_{k+q,k_1',k_2',k_2}) + (\Delta\epsilon_{k,k_1,k_2',k_2})}{(\Delta\epsilon_{k+q,k,k_1,k_1'})} \left( \frac{1 - \cos\left((\Delta\epsilon_{k+q,k,k_1,k_1'})t\right)}{(\Delta\epsilon_{k+q,k_1',k_2',k_2})^2 + (\Delta\epsilon_{k,k_1,k_2',k_2})^2} \right)
$$

$$
\times n_{k_1'}(n_{k_2'} - n_{k_2})\delta_{k+k_2',k_1+k_2}\delta_{k_1'-k_1,q}
$$

$$
+ \frac{U^2}{\Omega^4} \sum_k n_{k+q}(1 - n_k)
$$

$$
\times \sum_{k_i', k_i} \frac{1 - \cos\left((\Delta\epsilon_{k+q,k,k_1',k_1})t\right) - \cos\left((\Delta\epsilon_{k+q,k,k_2',k_2})t\right) + \cos\left((\Delta\epsilon_{k_1',k_1,k_2,k_2'})t\right)}{(\Delta\epsilon_{k+q,k,k_1',k_1})(\Delta\epsilon_{k+q,k,k_2',k_2})}
$$

$$
\times (n_{k_1'} - n_{k_1})(n_{k_2'} - n_{k_2})\delta_{k_1'-k_1,q}\delta_{k_2'-k_2,q}
$$

$$
- \frac{2U^2}{\Omega^4} \sum_k n_{k+q}
$$

$$
\times \sum_{k_i', k_i} \frac{1 - \cos\left((\Delta\epsilon_{k+q,k_1',k_2',k_2})t\right) - \cos\left((\Delta\epsilon_{k,k_1,k_2',k_2})t\right) + \cos\left((\Delta\epsilon_{k+q,k,k_1,k_1'})t\right)}{(\Delta\epsilon_{k+q,k_1',k_2',k_2})(\Delta\epsilon_{k,k_1,k_2',k_2})}
$$

$$
\times (1 - n_{k_1})n_{k_2'}(1 - n_{k_2})\delta_{k+k_2',k_1+k_2}\delta_{k_1'-k_1,q}
$$

$$
+ \frac{4U^2}{\Omega^4} \sum_k n_{k+q}
$$

$$
\times \sum_{k_1,k_2',k_2} \frac{1 - \cos\left((\Delta\epsilon_{k,k_1,k_2',k_2})t\right)}{(\Delta\epsilon_{k,k_1,k_2',k_2})^2}(1 - n_{k_1})n_{k_2'}(1 - n_{k_2})\delta_{k+k_2',k_1+k_2}
$$

$$
+ \mathcal{O}(U^3) . \tag{67}
$$

The calculation for the equilibrium correlation function is analogous. Now, we take the limit of infinite spatial dimensions [41]. This allows us to introduce energy integrals,

$$
\sum_{k_i} \dots \rightarrow \int \mathrm{d}\epsilon_i \sum_{k_i} \delta(\epsilon_i - \epsilon_{k_i})\dots , \tag{68}
$$

and make use of

$$
\sum_{k_1,k_2,k_3} \delta(\epsilon_1 - \epsilon_{k_1})\delta(\epsilon_2 - \epsilon_{k_2})\delta(\epsilon_3 - \epsilon_{k_3})\delta_{k+k_3,k_1+k_2}
$$

$$
\stackrel{d\rightarrow\infty}{=} \frac{1}{\Omega} \sum_{k_1,k_2,k_3} \delta(\epsilon_1 - \epsilon_{k_1})\delta(\epsilon_2 - \epsilon_{k_2})\delta(\epsilon_3 - \epsilon_{k_3}) \tag{69}
$$

and

$$
\sum_{k_1,k_2} \delta(\epsilon_1 - \epsilon_{k_1})\delta(\epsilon_2 - \epsilon_{k_2})\delta_{k_1-k_2,k} \stackrel{d\rightarrow\infty}{=} \frac{1}{\Omega} \sum_{k_1,k_2} \delta(\epsilon_1 - \epsilon_{k_1})\delta(\epsilon_2 - \epsilon_{k_2}) \qquad (\text{for } k \neq \vec{0}) . \tag{70}
$$

This means that from now on we restrict the domain of the Fourier transformed correlation function to values $q \neq \vec{0}$.

For the nonequilibrium correlation function, we get

$$
\hat{C}_q^{\uparrow\uparrow}(t) = \frac{1}{\Omega^2}\sum_k n_{k+q}(1-n_k)
$$

$$
-\frac{4U^2}{\Omega^2}\sum_k n_{k+q}(1-n_k)\int \mathrm{d}\epsilon_1 \int \mathrm{d}\epsilon_{2'}\int \mathrm{d}\epsilon_2 D(\epsilon_1)D(\epsilon_{2'})D(\epsilon_2)
$$

$$
\times\frac{1-\cos\left((\Delta\epsilon_{k,1,2',2})t\right)}{(\Delta\epsilon_{k,1,2',2})^2}\big((1-n_1)n_{2'}(1-n_2)+n_1(1-n_{2'})n_2\big)
$$

$$
+\frac{2U^2}{\Omega^4}\sum_k n_{k+q}(1-n_k)
$$

$$
\times\sum_{k_i',k_i}\frac{1-\cos\left((\Delta\epsilon_{k+q,k_1',k_2',k_2})t\right)-\cos\left((\Delta\epsilon_{k,k_1,k_2',k_2})t\right)+\cos\left((\Delta\epsilon_{k+q,k,k_1,k_1'})t\right)}{(\Delta\epsilon_{k+q,k_1',k_2',k_2})(\Delta\epsilon_{k,k_1,k_2',k_2})}
$$

$$
\times n_{k_2'}(1-n_{k_2})\delta_{k+k_2',k_1+k_2}\delta_{k_1'-k_1,q}
$$

$$
-\frac{2U^2}{\Omega^2}\sum_k n_{k+q}(1-n_k)\int \mathrm{d}\epsilon_{1'}\int \mathrm{d}\epsilon_1 \int \mathrm{d}\epsilon_{2'}\int \mathrm{d}\epsilon_2 D(\epsilon_{1'})D(\epsilon_1)D(\epsilon_{2'})D(\epsilon_2)
$$

$$
\times\frac{\Delta\epsilon_{k+q,k,2',2}}{\Delta\epsilon_{1',1,2,2'}}\left(\frac{1-\cos\left((\Delta\epsilon_{k+q,k,1',1})t\right)}{(\Delta\epsilon_{k+q,k,2',2})^2+(\Delta\epsilon_{1',1,2,2'})^2}-\frac{1-\cos\left((\Delta\epsilon_{k+q,k,2',2})t\right)}{(\Delta\epsilon_{k+q,k,2',2})^2}\right)
$$

$$
\times(n_{1'}-n_1)(n_{2'}-n_2)
$$

$$
+\frac{2U^2}{\Omega^4}\sum_k \big(n_{k+q}(1-n_k)+(1-n_{k+q})n_k\big)
$$

$$
\times\sum_{k_i',k_i}\frac{\Delta\epsilon_{k,k_1,k_2',k_2}}{\Delta\epsilon_{k+q,k_1',k_2',k_2}}\left(\frac{1-\cos\left((\Delta\epsilon_{k+q,k,k_1,k_1'})t\right)}{(\Delta\epsilon_{k+q,k_1',k_2',k_2})^2+(\Delta\epsilon_{k,k_1,k_2',k_2})^2}-\frac{1-\cos\left((\Delta\epsilon_{k,k_1,k_2',k_2})t\right)}{(\Delta\epsilon_{k,k_1,k_2',k_2})^2}\right)
$$

$$
\times n_{k_1}(n_{k_2'}-n_{k_2})\delta_{k+k_2',k_1+k_2}\delta_{k_1'-k_1,q}
$$

$$
+\frac{2U^2}{\Omega^2}\sum_k n_{k+q}(1-n_k)\int \mathrm{d}\epsilon_{1'}\int \mathrm{d}\epsilon_1 \int \mathrm{d}\epsilon_{2'}\int \mathrm{d}\epsilon_2 D(\epsilon_{1'})D(\epsilon_1)D(\epsilon_{2'})D(\epsilon_2)
$$

$$
\times\frac{(\Delta\epsilon_{1',1,2,2'})+(\Delta\epsilon_{k+q,k,2,2'})}{(\Delta\epsilon_{k+q,k,1,1'})}\left(\frac{1-\cos\left((\Delta\epsilon_{k+q,k,1,1'})t\right)}{(\Delta\epsilon_{1',1,2,2'})^2+(\Delta\epsilon_{k+q,k,2,2'})^2}\right)
$$

$$
\times(n_{k_1'}-n_{k_1})(n_{k_2'}-n_{k_2})
$$

$$
+\frac{2U^2}{\Omega^4}\sum_k \big(n_{k+q}(1-n_k)+(1-n_{k+q})n_k\big)
$$

$$
\times\sum_{k_i',k_i}\frac{(\Delta\epsilon_{k+q,k_1',k_2',k_2})+(\Delta\epsilon_{k,k_1,k_2',k_2})}{(\Delta\epsilon_{k+q,k,k_1,k_1'})}\left(\frac{1-\cos\left((\Delta\epsilon_{k+q,k,k_1,k_1'})t\right)}{(\Delta\epsilon_{k+q,k_1',k_2',k_2})^2+(\Delta\epsilon_{k,k_1,k_2',k_2})^2}\right)
$$

$$
\times n_{k_1'}(n_{k_2'}-n_{k_2})\delta_{k+k_2',k_1+k_2}\delta_{k_1'-k_1,q}
$$

$$
+ \frac{U^2}{\Omega^2} \sum_k n_{k+q}(1 - n_k) \int \mathrm{d}\epsilon_{1'} \int \mathrm{d}\epsilon_1 \int \mathrm{d}\epsilon_{2'} \int \mathrm{d}\epsilon_2 D(\epsilon_{1'}) D(\epsilon_1) D(\epsilon_{2'}) D(\epsilon_2)
$$

$$
\times \frac{1 - \cos\left((\Delta\epsilon_{k+q,k,1',1})t\right) - \cos\left((\Delta\epsilon_{k+q,k,2',2})t\right) + \cos\left((\Delta\epsilon_{1',1,2,2'})t\right)}{(\Delta\epsilon_{k+q,k,1',1})(\Delta\epsilon_{k+q,k,2',2})}
$$

$$
\times (n_{1'} - n_1)(n_{2'} - n_2)
$$

$$
- \frac{2U^2}{\Omega^4} \sum_k n_{k+q}
$$

$$
\times \sum_{k'_i, k_i} \frac{1 - \cos\left((\Delta\epsilon_{k+q,k'_1,k'_2,k_2})t\right) - \cos\left((\Delta\epsilon_{k,k_1,k'_2,k_2})t\right) + \cos\left((\Delta\epsilon_{k+q,k,k_1,k'_1})t\right)}{(\Delta\epsilon_{k+q,k'_1,k'_2,k_2})(\Delta\epsilon_{k,k_1,k'_2,k_2})}
$$

$$
\times (1 - n_{k_1}) n_{k'_2}(1 - n_{k_2}) \delta_{k+k'_2,k_1+k_2} \delta_{k'_1-k_1,q}
$$

$$
+ \frac{4U^2}{\Omega^2} \sum_k n_{k+q} \int \mathrm{d}\epsilon_1 \int \mathrm{d}\epsilon_{2'} \int \mathrm{d}\epsilon_2 D(\epsilon_1) D(\epsilon_{2'}) D(\epsilon_2)
$$

$$
\times \frac{1 - \cos\left((\Delta\epsilon_{k,1,2',2})t\right)}{(\Delta\epsilon_{k,1,2',2})^2}(1 - n_1) n_{2'}(1 - n_2)
$$

$$
+ \mathcal{O}(U^3) , \tag{71}
$$

with the density of state $D(\epsilon)$. With the assumption of zero temperature and the general ansatz $\epsilon_{k+q} \approx \epsilon_k + \nabla_k \epsilon_k \cdot q$, we can expand everything up to linear order in $q$ and arrive at the results in eqs. (40) and (41).

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
