# Peer review of "Prethermalization of density-density correlations after an interaction quench in the Hubbard model"

_SciPost Physics_

## Round 1 · Referee Report · Neil Robinson · 2019-9-22

Strengths

1- Detailed presentation with comprehensive appendices
2- Various consistency checks
3- Extends previous works from momentum space observables to real space correlation functions
4- Consistent with/supports the Burges et al. prethermalization picture

Weaknesses

1- Straightforward extension of previous work
2- Time-evolution step has to be approximated as free
3- Focus on prethermalization plateau (time-average) of observables
4- Referencing and context with previous works is lacking in some places

Report

In "Prethermalization of density-density correlations after an interaction quench in the Hubbard model" the authors use the flow equation approach to study non-equilibrium time-evolution of two-point correlation functions following a quench in the Fermi-Hubbard model (D>1). The authors consider quenches starting from the non-interacting part of the model at finite temperature. This is an extension of previous works by Moeckel and Kehrein [Phys. Rev. Lett. 100, 175702 (2008) and Ann. Phys. (N.Y.) 324, 2146 (2009)], which examined prethermalization of the mode occupation numbers following an interaction quench in the Fermi-Hubbard model with the same techniques, starting from a zero temperature initial state.

Section 2 of the paper recaps results from the previous works, performing the flow equations analysis for the time-evolution of the fermion creation/annihilation operators. These are pieced together in Sec. 3 to obtain the equal-time connected density-density correlation functions. The time-averages of these are compared compared to equilibrium correlation functions computed at the temperature of the initial state with the post-quench Hamiltonian. The main results are summarized in Eqs. (35) and (42): the prethermalization plateau result for the correlation function with antiparallel spins is, to leading order, the same as the said equilibrium thermal result. The correlation function with parallel spins has the same leading order long-distance asymptotic behavior as the equilibrium one.

The results of the manuscript with regards to the two-point functions are new, but are a simple extension of previous results. The authors discuss their main results, Eqs. (35) and (42), in the context of Berges et al: on the prethermalization plateau some observables (related to the proximate integrable constants of motion) retain memory of their initial conditions, while others (such as two point functions of local operators) rapidly lose memory of the initial conditions and equilibrate. In this regard, the results are in line with expectations.

The focus of discussion on the prethermalization plateau values, (35) and (42), is a bit of a missed opportunity with the full time-evolution at hand, (32) and (36). The work would be strengthened by discussing how the correlation functions approach their prethermal values and by providing a plot comparing the time-evolution of parallel/antiparallel-spin density-density correlation functions. This might also reveal some features not obvious from expressions (32) and particularly (36).

Overall, due to it being a straightforward extension of previous works and the results providing few new insights, I don't think the manuscript currently meets the criteria for publication in SciPost Physics.

There is also one point in the analysis which is not clear to me. To perform the time-evolution in the $B=\infty$ basis, the Hamiltonian (12) has to be approximated as non-interacting. Within the paper, this is justified by appeal to the interaction terms being those that appear in the quantum Boltzmann equation, and this is known to become relevant only at some late time scale, $t\sim \rho_F^{-3}U^{-4}$. This argument isn't convincing to me. If we consider the short-time expansion of the time-evolution operator, it is not clear to me why the elastic terms don't modify the short-time evolution of the density-density correlation function. In the case of a two fermion observables, such as the Green's function, the normal-ordering of the elastic collision term in (12) leads to an absence of $O(U)$ corrections, if I recall correctly. It's not obvious that the same is true for four-fermion operators, such as the density-density correlation function. An explicit calculation that illustrates this would strengthen the paper, and clarify this point.

Requested changes

1- At the end of Sec. 1.1, there is a discussion of the prethermalized post-quench state. If I understand correctly, at the order of calculations for which the results are presented, it would not be possible to distinguish between the discussed post-quench prethermal state (equilibrium values described by $\sim \exp(-H/T)$ with $T$ the initial state temperature and $H$ the full Hamiltonian) and the post-quench thermal state, whose temperature is presumably O(U) different. This should be clarified.

2- Following Eq. (4), Wegner's canonical generator is described as diagonalizing the Hamiltonian "apart from degeneracies". I think this phrasing is a little unclear -- the non-diagonal terms left in the transformed Hamiltonian are elastic collisions, but they don't only couple states at the same energies as implied by "degeneracies".

3- Add an explicit calculation showing that one can neglect the elastic collisions in the time-evolution for four-fermion operators.

4- In general, I found the referencing in the work lacking. An explicit example is Section 2.2: this is essentially the analysis of earlier works by Moeckel and Kehren (referenced above) which aren't cited here.

5- It would be good to clarify the difference between D>1 and D=1 for describing the prethermal value of correlation functions of local observables. For example, in this work the claim is that such correlation functions are already equilibrated to their thermal result (to leading order) on the prethermalization plateau, but my understanding of the D=1 case was that such observables could be described by an appropriate generalized Gibbs ensemble.

6- The results are a little overstated in the abstract. In the case of one of the density-density correlators, it is shown that prethermalized values match thermal ones. In the other, only the long-distance asymptotics are shown to match.

7- Some discussion of the real-time dynamics, including e.g. the approach to the prethermalization plateau and/or a plot of results for the time-evolution of the density-density correlations (for both parallel and antiparralel spins), would strengthen the paper and may help reveal interesting features that aren't obvious from the expressions (32), (36).

---

## Round 1 · Referee Report · Anonymous · 2019-11-24

Strengths

1- Interesting problem

Weaknesses

1- incremental extension of previous work;
2- I disagree with the authors' interpretation of the results (see report);

Report

The paper studies the non-equilibrium dynamics of the Fermi-Hubbard (FH) model in higher dimensions ($d>1$). The authors use the ``flow equation method" to perturbatively describe the dynamics of the FH model subject to a weak interaction quench. Namely, they prepare the system in the ground state of the FH Hamiltonian with no interaction and suddenly switch on a small interaction $U\ll 1$. The main idea of the paper is to compare the equal time density-density correlation function in the ``prethermalization regime" after the quench, with that on the thermal equilibrium state of the interacting model. This problem has been already considered by one of the authors in Ref. [9], which used the same techniques to study the behaviour of the momentum distribution.

In my view, apart from some minor issues discussed in the detailed points below, the paper suffers from two main weaknesses. First, the calculation presented here (although technically involved) represents a mere extension of those presented in Ref. [9]. Second, I disagree with the physical picture proposed by the authors. Indeed, the two ``definitions" of prethermalization used in the paper (that of Ref. [8] and that of Ref. [11]) are generically inequivalent: if one defines prethermalization as a ``near-integrability induced bottleneck in the thermalization dynamics" (definition attributed to Ref. [11]) it is absolutely not clear (and in general does not happen) that there exist some observables immediately attaining their thermal values as required by the definition of Ref. [8]. The current view is to consider the former (that of [11]) as the definition of prethermalization in condensed matter systems, see e.g.

A) F. H. L. Essler, S. Kehrein, S. R. Manmana, and N. J. Robinson, Phys. Rev. B 89, 165104 (2014),

coauthored by one of the authors of the current paper, and

B) B. Bertini, F. H. L. Essler, S. Groha, and N. J. Robinson, Phys. Rev. Lett. 115, 180601 (2015),

C) K. Mallayya and M. Rigol, Phys. Rev. Lett. 120, 070603 (2018),

D) P. Reimann and L. Dabelow, Phys. Rev. Lett. 122, 080603 (2019),

E) K. Mallayya, M. Rigol, and W. De Roeck, Phys. Rev. X 9, 021027 (2019),

as well as many of the references [12-23] cited in the paper. Specifically, one assumes that there exist an intermediate time window where the state in the expectation values of local(!) observables can be replaced by a ``deformed GGE" (in the sense of A), before they drift to their final thermal value. It appears to me that this description applies to the setting of the current paper, the only special feature here is that the quench injects an amount of energy density proportional to $U$, so that the occupations in the final thermal state will be order $U$ close those in the initial state. Therefore, I disagree with the interpretation of the authors that local quantities attain their final thermal value in the prethermalized regime. I think that they attain a non-thermal GGE value but, at the lowest non-trivial order in perturbation theory, the expectation values in this GGE are the same as the $T=0$ thermal ensemble. This is in agreement with the author's statement that the temperature will eventually raise: that value of the temperature (not 0) is the one of the final thermal ensemble.

I think that the authors should clarify this second point before the paper can be considered for publication.

Requested changes

1- I suggest to add the explicit definition of normal-ordered operators after Eq. 1

2- I suggest to rephrase the first paragraph of Sec. 2 to stress that all observables can be computed for for different initial states, not just density-density correlations.

3- I don't understand the sentence about the one-dimensional system after Eq. 4. Are the authors claiming that for the one-dimensional system the first and second term in Eq. 1 commute? If so I disagree. Please rephrase and clarify.

4- In the sentence about the interacting case after Eq. 6 I suggest to add that in the interacting case, even if in the energy-diagonal basis the form of the mode occupation is 6, the energy in the theta function is renormalised by the interactions.

5- What are ``higher order interaction terms" in Eqs. 7, 8, 15? Is it just a typo or the authors are referring to $n$-fermion terms with higher $n$?

6- Why do the authors evaluate the coefficients $\epsilon_k(B)$ and $V_{k_1',k_1,k_2',k_2}(B)$ up to the third order while $U_{k_1',k_1,k_2',k_2}(B)$ and the Hamiltonian only up to the second? Please give more explanation.

7- I am not sure I understand the discussion at in the beginning paragraph of section 3: to evaluate the equilibrium density-density correlation one has to use the ground-state occupations in Eq. 6 with $\epsilon_k$ replaced by the dressed dispersion relation, right? If so, please clarify the text.

8- In Eq. 33 there should be a restriction in the sum to avoid $\Delta \epsilon_{k,k',q',q}$ to vanish. The same applies to Eqs. 36 and 39.

9- In the discussion after Eq. 39, when mentioning that they can prove that the density-density correlations in the prethermal regime coincide with those at equilibrium, the authors should immediately say that this holds for $d\to \infty$.

10- In the second part of Sec. 3.4 the authors talk about a finite temperature $T$, but it seems to me that their calculations are at $T=0$.

---

## Editorial Decision

editor-in-charge_assigned